# Channelized topography amplifies melt-sensitivity of cold Antarctic ice shelves

Qin Zhou ®[1,8] ✉, Tore Hattermann ®[2,8] ✉, Chen Zhao ®[3], Rupert Gladstone ®[4], Julius Lauber[2,5], Petteri Uotila ®[6] & Ashley Morris ®[7]

The stability of Antarctic ice shelves, which regulate the flow of grounded ice into the ocean, depends critically on ocean-driven basal melting. Basal channels, widespread features beneath many ice shelves, modulate ice-shelf basal melt rates and influence ice-shelf stability, yet their oceanic drivers remain poorly understood. Using high-resolution simulations of a cold-water ice shelf cavity, we show that interactions between circulation and channelized topography generate localized overturning that traps intruding warm Circumpolar Deep Water (CDW) beneath the ice, amplifying melt rates by an order of magnitude within channels. This ocean-driven process significantly enhances the sensitivity of the ice shelf basal mass loss to ocean warming, and the resulting differential melting promotes channel growth, with the potential to undermine the structural stability of the deeper part of the ice shelf. Our results reveal a key mechanism for basal channel evolution and indicate that even modest CDW intrusions could have important implications for the stability of cold Antarctic ice shelves.

The Antarctic Ice Sheet (AIS) stores 70% of the Earth's freshwater and is the largest potential contributor to sea level rise in a warming climate[1]. While global sea level is predicted to rise by several tens of centimeters by the end of the 21st century, a rise of up to 5 m by 2150 under a high-emission scenario cannot be ruled out due to the large uncertainty in the AIS[2,3]. The fate of the AIS is dynamically linked to the integrity of its floating ice shelves overlying vast ocean cavities. By exerting resistive stress, these ice shelves buttress the grounded ice and regulate its discharge into the ocean. Ocean-driven thinning of ice shelves has considerably reduced buttressing and is now the primary driver of AIS mass loss in recent decades[4,5]. However, processes governing the ocean-driven thinning remain poorly constrained, leading to uncertainty in sea-level rise projections[6]. One such uncertainty arises from basal channels − elongated troughs, several kilometers wide and up to hundreds of meters deep, incised into the undersides of many Antarctic ice shelves[7–9]. They can persist for decades and traverse for

hundreds of kilometers from the grounding line toward the ice front[8,10,11]. By redistributing basal melting, basal channels influence ice-shelf basal melt rates and spatial patterns, an important factor for ice-shelf stability. Despite increasing observational and modeling efforts, it remains unclear whether basal channels stabilize or destabilize Antarctic ice shelves. The former could be facilitated by preventing area-wide basal melting, leading to a net strengthening of ice shelves[12,13]. Alternatively, basal channels concentrate melting[14] and promote structural weakening[10,15], potentially leading to ice-shelf destabilization[16].

Basal channels can be initiated mechanically by undulations in basal topography at the grounding line, or in association with velocity gradients at ice shelf shear margins[12,17,18]. In addition, outflows of meltwater from beneath the ice sheet enhance channelized melting near ice shelf grounding lines[19–22]. From an oceanic perspective, in the absence of upstream ice-dynamical or subglacial forcing, sustaining

[1]Akvaplan-niva, Tromsø, Norway. [2]Norwegian Polar Institute, Tromsø, Norway. [3]Australian Centre for Excellence in Antarctic Science & Australian Antarctic Program Partnership, Institute for Marine and Antarctic Studies, University of Tasmania, Hobart, Australia. [4]Arctic Centre, University of Lapland, Rovaniemi, Finland. [5]Multiconsult, Tromsø, Norway. [6]Institute for Atmospheric and Earth System Research/Physics, University of Helsinki, Helsinki, Finland. [7]Svalbard Integrated Arctic Earth Observing System, Longyearbyen, Norway. [8]These authors contributed equally: Qin Zhou, Tore Hattermann. ✉e-mail: qin.zhou@akvaplan.niva.no; tore.hattermann@npolar.no

channelized morphology against ice creep closure[23–25] requires differential melting, that is, higher melt rates at the shallower channel crests and lower melt rates at the channel base (the deepest part of the channel)[17]. However, the thermodynamics that govern basal melting through the difference between in-situ temperature near the ice base and the local pressure-dependent freezing point, referred to as thermal driving hereafter[26], suggests the opposite. Both a higher freezing point and the accumulation of colder, buoyant meltwater near the channel crest reduce thermal driving, thereby suppressing rather than enhancing basal melting. Simplified plume models suggest that basal channels may spontaneously form through localized enhanced melting associated with spatial inhomogeneities and grow through entrainment of warmer water into the meltwater plume[12,17]. But such models only partially resolve the circulation inside the channel, while other studies suggest that buoyant plumes enhance melting primarily along the Coriolis-favored sloping flank, rather than the top of the channel[27], driving an asymmetric melt pattern that can displace the channel laterally, but does not necessarily contribute to its vertical deepening[11].

Furthermore, these mechanisms require relatively warm ocean conditions to produce basal melt rates that are large enough to dominate the ice deformation[17], which has a tendency to close the channels by leveling out the ice thickness gradients through ice-creep[15,23,24]. These conditions are typically met in West Antarctica[10,16], where large amounts of Circumpolar Deep Water (CDW) with temperatures of several centigrade above the local melting point inside the ice-shelf cavities[28] drive melt rates of several tens of meters per year[29,30]. Still, basal channels are also abundantly observed beneath low-melting ice shelves in East Antarctica[8,31], where access of CDW is limited or absent, and cavities are primarily filled with colder water at or near the freezing point[32]. Hence, a consistent understanding of the oceanic processes that govern the evolution of basal channels is still lacking.

One such cold-water ice shelf cavity exists below the Fimbulisen Ice Shelf, which is located around the Prime Meridian in the Atlantic sector of the Southern Ocean. Ground-penetrating radar data[31] and the fine-resolution (8 m) Reference Elevation Model of Antarctica (REMA, Materials and Methods) reveal channelized basal topography extending tens of kilometers along and across the central Jutulstraumen ice stream. The oceanographic setting at Fimbulisen is characteristic of the "fresh shelf" regime[32], where a pronounced Antarctic Slope Front separates CDW (or its regional derivative, Warm Deep Water; see Materials and Methods) from the continental shelf. On the continental shelf, less saline Winter Water (WW), with temperatures near the freezing point, results in low average basal melt rates[33–35]. However, occasionally, traces of CDW reach the ice shelf cavity[33,36,37], and observations from a long-term moored observatory below Fimbulisen show that CDW intrusions strengthened after 2016, and might continue to do so in the future[35].

In this study, we use a high-resolution ice shelf-ocean model to show how the CDW intrusions below Fimbulisen interact with colder cavity waters to create self-organizing patterns of differential melting that promote channel formation beneath deeper parts of the ice shelf. Our results reveal a novel oceanic mechanism in which the trapping of buoyant warmer water inside the channels enhances local thinning, suggesting that even moderate CDW intrusions may affect the stability of ice shelves with cold and fresh cavities.

## Results

### Modeling the effects of basal channels in a cold-water ice shelf cavity

To investigate the interaction of the observed basal morphology with the varying oceanic forcing at Fimbulisen, we conducted four numerical simulations using the Finite Volume Community Ocean Model (FVCOM)[38,39], configured with a locally refined unstructured grid at

50 m resolution within the cavity to resolve small-scale basal features ("Methods"). The experiments combine either SMOOTH or ROUGH ice draft geometries, representing the absence or presence of basal channels ("Methods", Fig. S1), with COLD or WARM ocean forcing, corresponding to the absence or presence of moderate CDW intrusions (Fig. 1c, d). The general circulation in our simulation with COLD forcing and ROUGH draft resembles results from previous process-oriented Fimbulisen modeling[34] (Fig. S2), with ocean temperatures close the surface freezing point (Fig. 1c) inside the cavity and relatively low cavity-averaged melt rate of 0.5 m yr⁻¹. The basal melt pattern (Fig. 1a) agrees well with other estimates from models and observations[31,34,40]. Enhanced melting of several meters per year occurs below the deepest part of the Jutulstraumen keel, and at the Trolltunga ice tongue that overhangs the continental shelf break[31], while much of the flatter central part of the ice shelf exhibits low melting of less than one meter per year and occasional refreezing.

In the region that resolves the small-scale basal morphology in the ROUGH ice draft geometry (cyan polygon in Fig. 1a), the presence of basal channels imprints on the melt rate distribution and on the spatial pattern of melt rate differences between the ROUGH and SMOOTH draft experiments (Fig. S3), in particular below the southern Jutulstraumen, where channels with amplitudes of hundred meters are carved into the thicker keel of the ice shelf (Fig. 1e). Large Rossby numbers in these areas (Fig. S4) support recent findings that the small-scale basal topography promotes the formation of submesoscale eddies that enhance vertical heat transport to the ice-shelf base[41]. Downstream of this region, melt-rate differences are small due to the diminishing ice draft difference (Fig. S1c). Consistently, the area-averaged basal melt rates are higher under the ROUGH ice draft than under the SMOOTH ice draft, increasing by 18% (27%) in the smoothing region and 21% (35%) in the deep-ice region under COLD (WARM) forcing conditions (Fig. 1b, "Methods"). This increase reflects a spatially consistent enhancement of grid-point melt rates across the ice shelf (Wilcoxon signed-rank test, $p < 0.001$, "Methods").

### CDW intrusion amplifies channelized melting

Under WARM ocean forcing, CDW enters over a sill in the seafloor on the western side of Trolltunga[42] and propagates along the topographic contours further into the cavity (Fig. 1d), consistent with observed warm inflow events at nearby ice shelf cavity moorings[37]. While the WARM forcing enhances basal melting below the entire ice shelf (Fig. S3c), the cavity-averaged melt rate (-1 m yr⁻¹) remains relatively low compared to West Antarctic ice shelves (-10 m yr⁻¹)[40,43], which are exposed to much larger amounts of CDW inside their cavities[44]. However, the presence of basal channels significantly enhances the melting increase due to the CDW intrusions (Fig. 1b). Especially in the deep-ice region, the increase in area-averaged melt rates from COLD to WARM forcing is four times larger under the ROUGH ice draft (0.4 m yr⁻¹) compared to the SMOOTH ice draft (0.1 m yr⁻¹). We find that this response is associated with disproportionally large melt rates inside the channels along the deep Jutulstraumen keel, where melt rate anomalies between the ROUGH and SMOOTH draft exceed 10 m yr⁻¹ under WARM forcing (Fig. 2b, d).

A cross-section of a 150 m high and 3 km wide channel on the western side of the Jutulstraumen keel (Fig. 1a) exemplifies how the channelized flow amplifies the melt rate response when CDW is present in the cavity (Fig. 2). Under the ROUGH ice draft, a buoyant meltwater-laden plume establishes along the western side of the channel, accompanied by a weaker return flow on the eastern side (Fig. 1e, f). Since the turbulent heat transfer in the applied basal melt parameterization is proportional to the ocean velocity near the ice base[45] ("Methods"), the flow speed anomalies of around -0.2 m s⁻¹ associated with these channelized buoyant outflows (Fig. 2e, f) drive positive melt rate anomalies of several meters per year along the western flank of the channel (Fig. 2c, d). While the ocean speed is

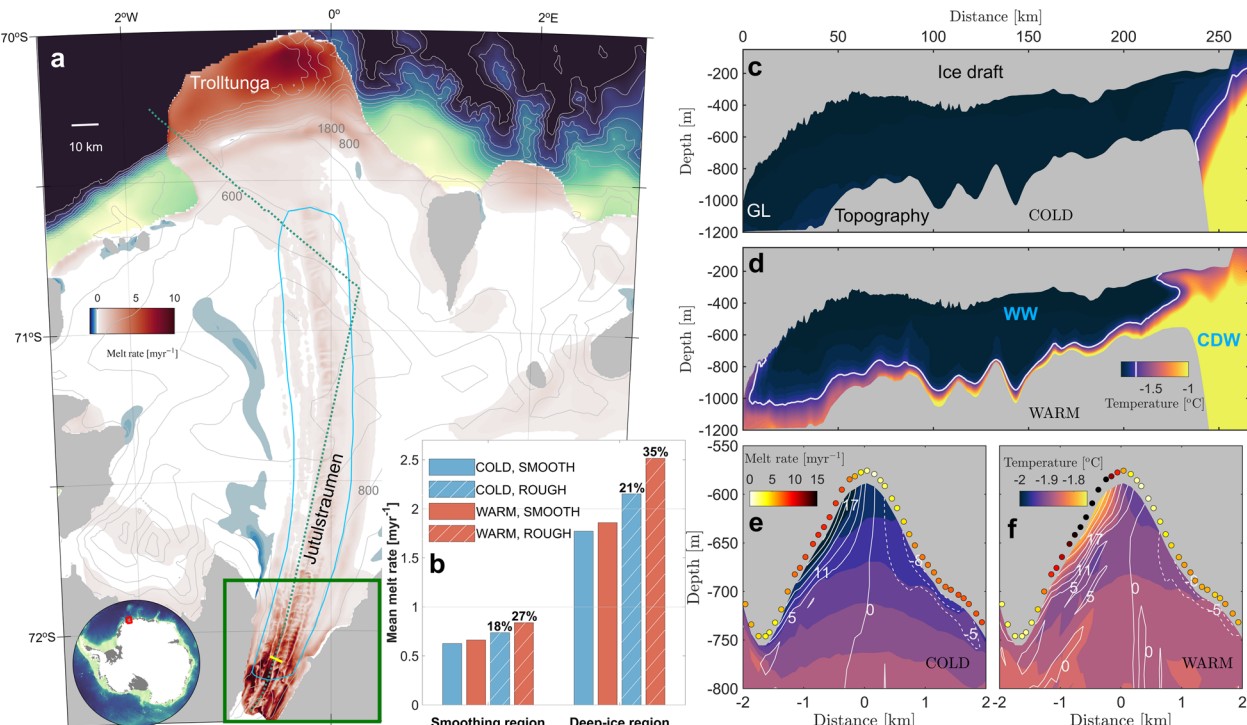

**Fig. 1 | Overview of Fimbulisen Ice Shelf and main findings. a** Map of the Fimbulisen Ice Shelf showing simulated basal melt rates for the present Fimbulisen and bottom bathymetry (thin gray lines). The cyan polygon outlines the smoothing region, where basal channels are selectively removed in the SMOOTH draft experiments. The long, thin green dashed line indicates the cross-domain transect illustrating COLD and WARM ocean forcing in **c**, **d**. The short, thick yellow line indicates a cross-section of an example basal channel illustrated in **e**, **f**. The green box indicates the deep-ice region used for focused analysis. **b** Averaged melt rates over the smoothing and deep-ice regions for both ROUGH and SMOOTH ice draft geometries under COLD and WARM forcing. Melt rate increases (%) in the ROUGH draft experiments relative to the SMOOTH draft are indicated above the bars. **c**, **d** Temperature cross-sections along the cross-domain transect under COLD (WARM) ocean forcing. White lines indicate the isothermal line of −1.7 °C, denoting the boundary between CDW and WW qualitatively. GL denotes grounding line. Cross-channel (west to east) temperature sections through the example channel under COLD (**e**) and WARM (**f**) forcing. White solid (dashed) lines indicate along-channel outflows (return flows) in cm s⁻¹. Colored-filled dots indicate melt rates along the cross-section.

enhanced along the channel under both COLD and WARM ocean forcing, anomalies in thermal driving, another factor in the basal melt parameterization ("Methods"), differ remarkably depending on whether CDW is present inside the cavity or not. Under COLD forcing, thermal driving anomalies between the SMOOTH and the ROUGH ice draft range from ~+0.15 °C at the channel base to below ~−0.05 °C near the crest of the channel. This asymmetric pattern reflects the combined effect of the pressure-dependent freezing point, which increases by ~+0.1 °C between the channel base and the channel crest, and the trapping of cold and buoyant Ice Shelf Water (<−2.0 °C; Fig. 1e) that rises to the top of the channel. The reduced thermal driving partially offsets the effect of the enhanced ocean speed, such that maximum melt rate anomalies of a few meters per year (relative to the SMOOTH draft) occur along the western side of the channel under COLD forcing (Fig. 2c). Under WARM forcing, in contrast, warmer water (>−1.7 °C) appears inside the channel (Fig. 1f), causing positive thermal driving anomalies (relative to the SMOOTH topography) of ~+0.15 °C from the channel base until 0.3 km upstream of the apex. While the topographically enhanced friction velocity generally increases melting under the ROUGH ice draft, this alternate thermal structure under WARM forcing additionally amplifies the effect of the enhanced ocean speed inside the channel (Fig. 2f), leading to pronounced melt rate anomalies of more than ten meters per year near the crest of the channel when CDW is present in the cavity (Fig. 2d).

### Melt-driven overturning of CDW inside basal channels
The interactions of WW and CDW with the ice shelf lead to a self-organizing pattern where the warmest water is trapped at the top of the channels. Being more saline, and hence denser (potential density

~27.8 kg m⁻³; Fig. 3a), the CDW that enters below the ice shelf settles beneath more buoyant but colder WW to flow along the seafloor into the cavity (Fig. 1d). Along their path, mixing with WW transforms the warm inflows into modified CDW (mCDW, Fig. 3a), while the stratification with WW is preserved (Fig. 1d), allowing the mCDW to interact with the ice base upon reaching the deeper part of the cavity. The cooling and freshening from basal melting further transforms the mCDW properties along the meltwater-mixing line[46] ("Methods"), lowering its potential density from above ~27.7 kg m⁻³ to below ~27.6 kg m⁻³ (mCDW′, Fig. 3a). While the mCDW becomes increasingly buoyant as it melts the ice shelf, the WW follows a separate meltwater-mixing, which allows only for a limited decrease in density from ~27.66 kg m⁻³ to ~27.61 kg m⁻³, before the WW has cooled to the local melting point and depleted all its melting potential (WW′, Fig. 3a). Being warmer (~>0.4 °C) and more buoyant, the meltwater-modified mCDW rises above the meltwater-modified WW, forming a localized density-driven overturning near the grounding line (Fig. 1d).

The widespread channeling of mCDW under the ROUGH ice draft is evident from the spatial distribution of the reference salinity, that is used to differentiate the origin of different water masses below the ice base (defined as the intersect of the meltwater mixing line of a specific water mass with the surface freezing temperature, see "Methods" for details). Reference salinities of around 34.2 that mark the warmest water at the crest of the example channel (Fig. 2d) are tracing most of the channels along the deep Jutulstraumen keel (Fig. 3c). This topographic confinement of the overturned meltwater-modified mCDW explains why the basal melting is particularly enhanced inside these channels when CDW is present in the cavity. Under the SMOOTH draft, in contrast, smoothing removes the basal channels along the keel and

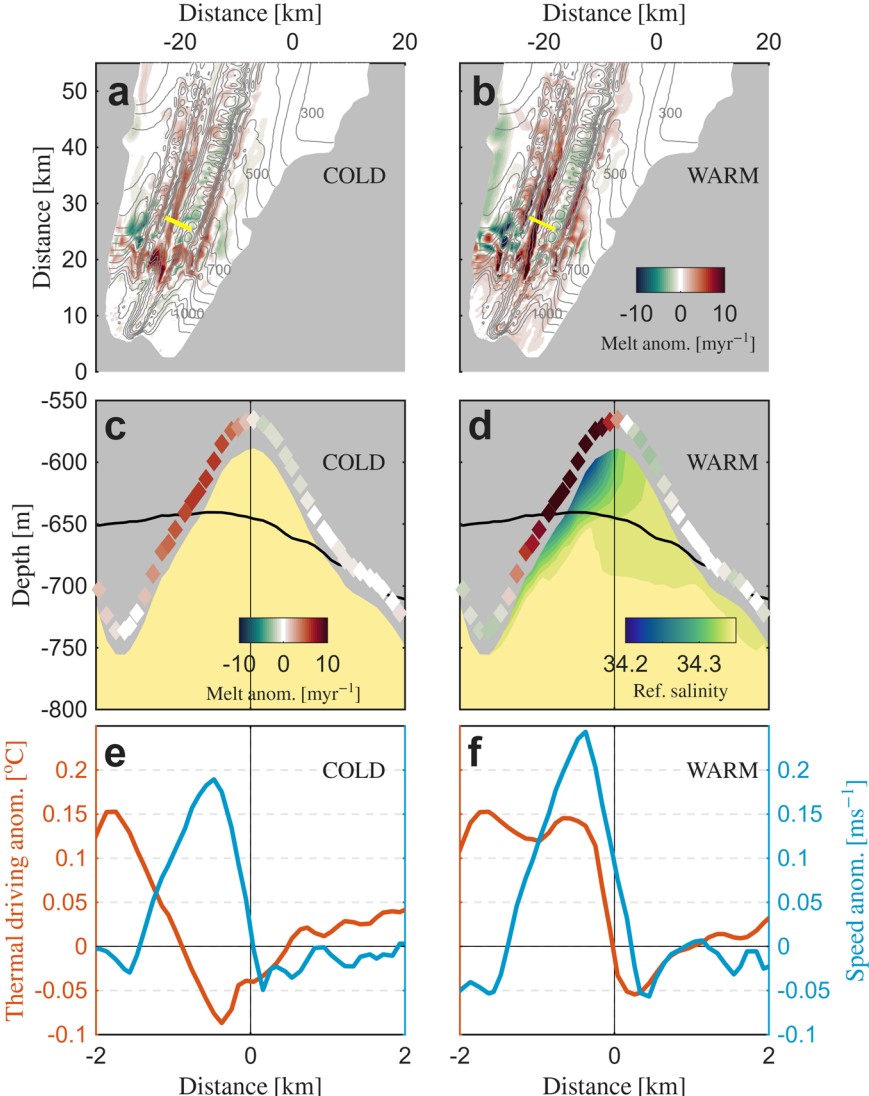

**Fig. 2 | Anomalies in melt rates, thermal driving and flow speed in the ROUGH draft experiments relative to the SMOOTH draft.** Spatial distribution of melt rate anomalies under COLD (**a**) and WARM (**b**) forcing. The yellow solid line indicates the cross-channel (west to east) sections through the example channel illustrated in the following panels. Gray contours indicate the ROUGH ice draft at 50 m intervals. Reference salinity cross-sections and melt rate anomalies (color shading diamonds) along the cross-section under COLD (**c**) and WARM (**d**) forcing. Black solid lines represent the smoothed version of the channel in the SMOOTH draft. Thermal driving anomalies (in red) and flow speed anomalies (in blue) along the cross-section under COLD (**e**) and WARM (**f**) forcing.

thus the topographically guided along-channel flow. Consequently, the meltwater modified mCDW is diverted by flow along the ice base and advected towards the shallower flanks of the ice shelf (Fig. S5b), leaving nearly no trace of modified mCDW in the smoothing region along the Jutulstraumen keel (Figs. 2c and 3b).

## Discussion

Enhanced melting within the basal channels, driven by the trapped meltwater-modified CDW, results in differential basal melting across the channel. To assess the impact of this differential melting on the channel evolution, we compare basal melt rates across the example channel (Fig. 4) with ice-dynamic thinning rates derived from a stand-alone ice-dynamic model ("Methods"). This comparison suggests an instrumental role of the occasional inflows of CDW below Fimbulisen in maintaining the observed channelized geometry near the Jutulstraumen keel. Ice dynamics alone promote channel closure, as creep induces thinning at the channel base and thickening at the crest (Fig. 4a). Our analysis shows a spatial thinning rate anomaly due to this effect rather than absolute values ("Methods"). Under

COLD forcing, the combined effect of ice dynamics and ocean melting yields consistently lower thinning rates inside than outside the channel (solid blue curve, Fig. 4b), indicating net channel closure. Under WARM forcing, the combined thinning rates on the western flank near the crest of the channel are 5–10 m yr$^{-1}$ higher compared to the thinning rates outside the channel. This suggests that basal channels can be sustained, or even grow, with mixed CDW and WW water masses in the cavity, but tend to close in the absence of CDW.

Recent advances in spatially refined satellite-derived estimates of basal melt rates may help identify patterns of differential melting[47]. However, uncertainties of several meters per year remain large relative to the expected signals beneath moderately melting ice shelves such as Fimbulisen[35,40]. Repeated Digital Elevation Models (DEMs) may reveal channel migration and evolution[48], but this approach requires disentangling ice-advection effects from melt-driven thickness changes at individual channels on interannual timescales. Ultimately, ground-based, process-oriented observations, such as time series from autonomous phase-sensitive radio-echo sounders[49], can provide

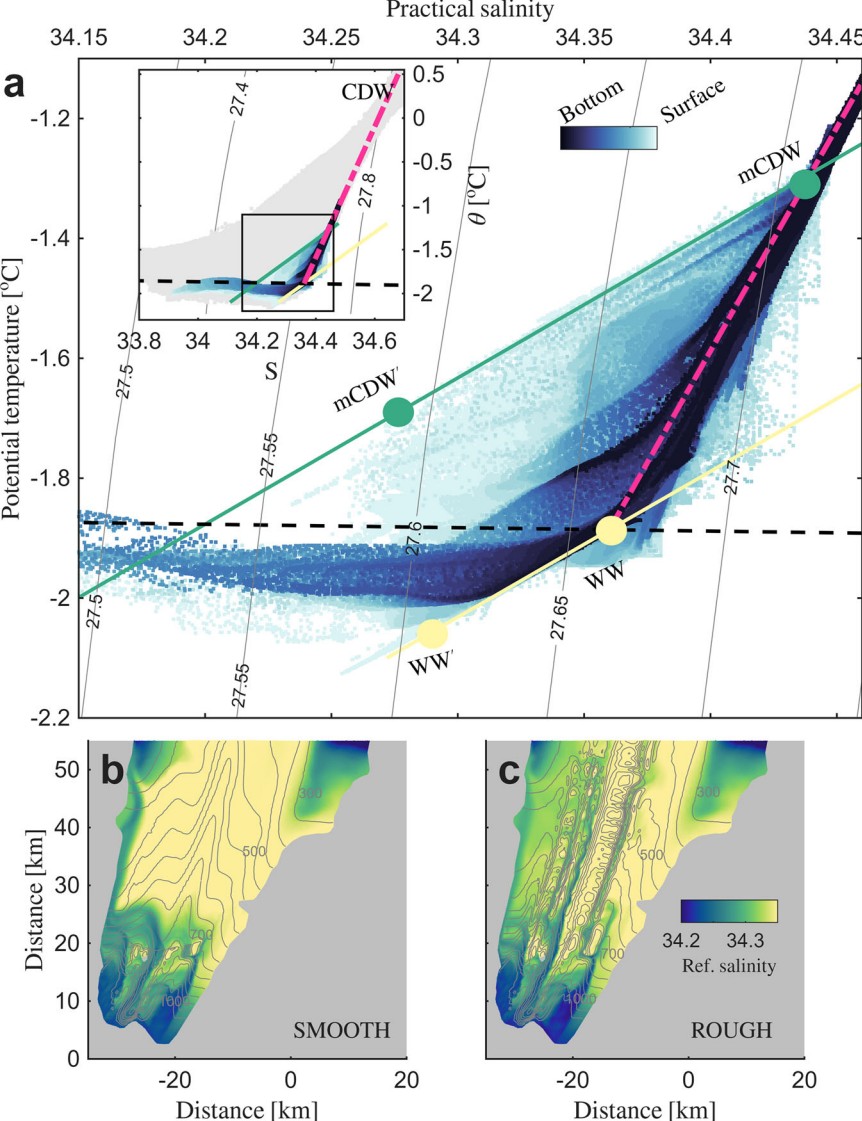

**Fig. 3 | Water mass properties in the deep-ice region in the WARM forcing experiments. a** Temperature-salinity diagram, showing water properties of modified Circumpolar Deep Water (mCDW) and Winter Water (WW) in the deep-ice region, with dark (light) blue indicating water located at the bottom (surface). The pink dashed line indicates the CDW-WW mixing line, and the black dashed line marks the surface freezing point referenced by surface pressure. Isopycnals are shown as gray curved lines. The green (yellow) solid line represents the meltwater- mixing line of mCDW (WW). The two green (yellow) dots along this line mark the transformation from source water (upper dot) to the meltwater-modified version, mCDW′ (WW′) (lower dot), due to ice-ocean interaction. The inset shows water mass properties in the deep-ice region relative to the full domain (light grey points). Spatial distributions of reference salinity at the ice-ocean interface in the SMOOTH (**b**) and ROUGH (**c**) draft experiments, with blue (yellow) color indicating mCDW (WW) sourced water and gray contours showing the ice draft at 50 m intervals.

accurate reference measurements, but require dedicated field deployments and resources.

Channelized melting beneath colder ice shelves such as Fimbu- lisen has previously often been linked to subglacial discharge[19,50], and we assume that similar processes are also at play at the Jutulstrau- ment grounding line[51]. However, the simulations presented in this study provide, to our knowledge, the first explicit example of an ocean-driven process that may promote basal channel growth beneath a slowly melting ice shelf through differential melting, which has previously been attributed to warm water cavities[17]. The key ingredient in this mechanism is the coexistence of distinct water masses, with warmer, initially denser CDW underlying colder, lighter WW. Because their properties evolve along separate meltwater- mixing lines, CDW can become buoyant while remaining warm, generating a localized density-driven overturning that causes war- mer water to be trapped inside the channels. In principle, these

dynamics work in any ice shelf cavity, where CDW is overlain by lighter and colder WW.

The time-averaged forcing with COLD and WARM conditions in our experiments is chosen to clearly delineate the effect of the CDW intrusions that are observed to occur in pulses below Fimbulisen[35]. Despite this idealization, the magnitude and vertical extent of the near- bottom temperature increase under the WARM forcing are consistent with ice-shelf borehole observations[37,52]. If such temperature anoma- lies propagate farther into the cavity, as inferred by ref.[53], episodic warm inflows may therefore contribute to maintaining the channelized geometry observed at the Jutulstraumen ice stream.

Furthermore, recent oceanic observations indicate increased access of CDW along the East Antarctic coast[35,54–57], and model pro- jections suggest that these trends might continue into the future[35,58–60]. Our simulations show that the melt-sensitivity to such relatively subtle changes–compared to the more dramatic shift from a cold to a warm

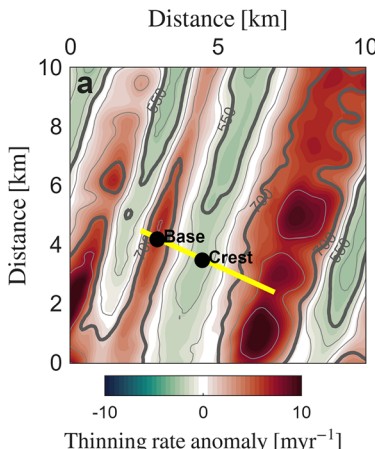
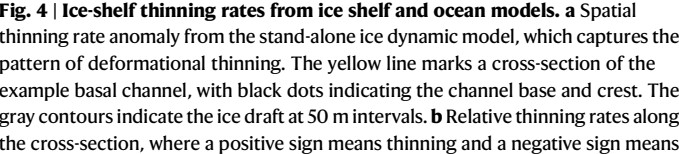
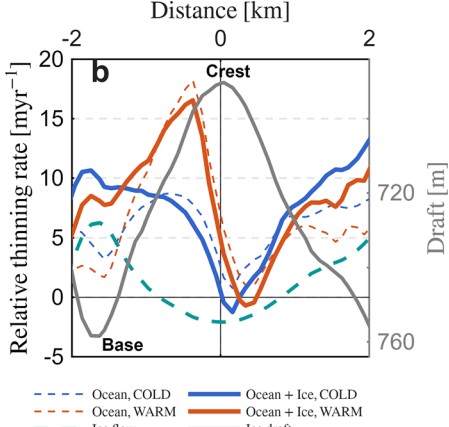

**Fig. 4 | Ice-shelf thinning rates from ice shelf and ocean models. a** Spatial thinning rate anomaly from the stand-alone ice dynamic model, which captures the pattern of deformational thinning. The yellow line marks a cross-section of the example basal channel, with black dots indicating the channel base and crest. The gray contours indicate the ice draft at 50 m intervals. **b** Relative thinning rates along the cross-section, where a positive sign means thinning and a negative sign means thickening. Blue and red dashed lines represent ocean-driven ice-shelf thinning rates under COLD and WARM forcing, respectively. The turquoise dashed line shows the thinning rate anomaly from the stand-alone ice-flow model. Solid lines indicate the combined ocean-driven and ice-dynamic thinning rate under COLD (blue) and WARM (red) forcing. The grey solid line shows the ice draft along the cross-section.

cavity as proposed for the Filchner–Ronne Ice Shelf[61]–can be strongly amplified through interactions of ocean circulation with small-scale channelized topography. In this way, even modest CDW incursions can significantly increase basal melt rates below the deep ice, promote channel growth, and potentially compromise ice shelf stability near the grounding line. The novel channelized overturning mechanism described in this study suggests that weakly melting ice shelves may be more vulnerable to moderate CDW intrusions than previously thought.

Overall, our findings show small-scale basal topography can augment the area average basal melting by promoting vertical heat transfer to the ice base[41], rather than stabilizing ice shelves as was suggested by earlier idealized modeling[12,13]. Omitting the effects of small-scale basal topography substantially alters the simulated melt patterns, highlighting that the evolution of the coupled ice sheet-ocean system cannot be assessed without accounting for the effects of channelized melting that impact the dynamical thinning and ice shelf buttressing near the grounding line. Incorporating the effect of these dynamics into coupled ice sheet-ocean models is an important but difficult challenge for reducing uncertainties in Antarctic mass balance estimates and future sea-level rise projections.

## Methods
### High-resolution REMA ice draft
We derived high-resolution (8 m) ice draft topography from the REMA dataset[62], which was generated from 2 m resolution stereo WorldView satellite images using photogrammetry. To minimize artifacts caused by ice flow in the standard REMA mosaic, we constructed a new DEM for the Fimbulisen Ice Shelf using individual REMA DEM strips. These strips were corrected for ice flow using ice velocity data from ITS_LIVE[63,64] and vertically adjusted to minimize the offset to CryoSat-2 altimetry[65,66]. The resulting mosaic preserves small-scale topographic features while aligning with large-scale surface geometry. Freeboard was calculated by subtracting the EIGEN-6C4 geoid[67] and a mean dynamic topography of −1.3 m[68], and converted into ice thickness and draft using standard hydrostatic equations, accounting for firn air content[69].

### Weddell Gyre derivative of CDW
In the Atlantic sector of the Southern Ocean, pure CDW at temperatures of around 2 °C is found in the Antarctic Circumpolar Current. Mixing with colder surrounding waters along the advective pathway

through the Weddell Gyre reduces the sub-surface temperature maximum of this water mass to about 1 °C at the continental slope off Fimbulisen[70]. In a regional context, this water mass is also referred to as Warm Deep Water[71], while we adopt the label CDW for simplicity, referring to the potential sub-surface heat source for ice shelf melting around Antarctica.

### Fimbulisen ice shelf-ocean model
The Fimbulisen ice shelf-ocean model uses the unstructured FVCOM that can resolve ice shelf-ocean interactions[38,39]. In-situ density is calculated using the polynomial equation of state[72], as implemented in FVCOM, with pressure explicitly included. The domain spans from 4°W to 5°E and from the grounding line at about 71.8°S to the open ocean at about 68.9°S, covering the Fimbulisen cavity and the Antarctic Slope Current carrying CDW at the shelf break. The horizontal grid resolution varies from 50 m in deep-ice regions to resolve small-scale basal features to 500 m over the continental slope to resolve mesoscale eddies and up to 1500 m in the open ocean. The model is discretized into 24 evenly spaced terrain-following layers in the vertical. Bathymetry is based on the RTopo2 data set[73], with updated data below Fimbulisen[74].

The ice draft used in the ROUGH draft simulations (Fig. S1a) is derived from the high-resolution REMA ice draft data, and the draft in the SMOOTH draft simulations is a smoothed version of the ROUGH draft (Fig. S1b). We applied smoothing with the Gridfit surface fitting algorithm[75] mainly along the Jutulstraumen keel, roughly confined by the 300 m draft contour (Fig. S1a). By setting the smoothness parameters to 0.3 at the along-ice-stream direction and one at the cross-ice stream direction, we aimed to smooth out basal channels aligned with the ice stream. The region within 20 km of the grounding line is excluded from smoothing to preserve the true depth of the deepest ice and to avoid violations of the minimum water column thickness criterion in the ice shelf-ocean model with terrain-following coordinates[76]. Smoothing near the grounding line could lead to adjustments to the water column thickness to satisfy this criterion, causing artificial changes in circulation and melt rates. By excluding this region, we ensure that the effects of small-scale basal features remain physically realistic and discernible. Outside the smoothing region, both ROUGH and SMOOTH drafts are smoothed uniformly in all directions at a 240 m resolution using the Gridfit algorithm. After smoothing, the cross-stream-averaged drafts of the SMOOTH and

ROUGH versions show negligible differences (Fig. S1c). However, small-scale basal features with depths in the order of 100 m are largely removed in the SMOOTH draft (Fig. S1d, e). At the southern boundary of the smoothing region, Root Mean Square Deviation (RMSD) of drafts relative to the mean draft within a 3 km bin along the ice stream − a draft roughness metric − decreases from 80 m in the ROUGH draft to 40 m in the SMOOTH draft. This difference diminishes downstream and stabilizes at approximately 3 m beyond 50 km from the southern smoothing boundary (Fig. S1c). We therefore define the area from the grounding line to 50 km downstream of the southern smoothing boundary as the deep-ice region, where roughness differences between the SMOOTH and ROUGH drafts are most pronounced. This region is the focus of our model analysis.

Lateral boundary conditions were taken from the eddy-resolving Fimbulisen ice shelf-ocean model based on the Regional Ocean Model System[34], which has been used to investigate basal melting and cavity circulation beneath Fimbulisen under different CDW inflow scenarios that have been observed[35]. The variability of CDW inflow is controlled by the wind stress and hydrographic forcing, with experiment names reflecting combinations of winter or summer hydrography and weak (30), mean (100), and strong (130) wind forcing. From the 18 scenarios, we selected the WIN-100 and SUM-30 climatologies as lateral boundary conditions for the COLD and WARM forcing experiments, respectively. WIN-100 represents a scenario with a homogeneous WW-filled cavity and no CDW inflow, and SUM-30 represents a scenario with moderate CDW inflow to the cavity, resulting in the highest melt rates in the deep-ice region. All simulations were initialized from the WIN-100 climatology. In the COLD forcing experiments, cavity conditions consistent with WIN-100 hydrography and circulation were maintained through the lateral open-boundary conditions. In the WARM experiments, the cavity transitions from the initial COLD state to WARM conditions through lateral open-boundary forcing derived from the SUM-30 climatology.

The air-sea boundary conditions were simplified by covering the open ocean with a one-meter-thick artificial ice draft. In our process-oriented simulations, the presence of CDW within the cavity is primarily determined by the depth of the Antarctic Slope Front, which is prescribed through the lateral boundary conditions. Although the artificial ice layer modifies surface fluxes, such as air-sea heat and freshwater exchange, it does not impact the water masses entering the cavity during the simulation period (less than 1 year).

Although the local vertical spacing of sigma layers under the ROUGH and SMOOTH drafts differs, the resulting changes in surface-layer sampling depth have only a minor influence on basal melt rates in our simulations, which are instead dominated by circulation changes associated with the channel geometry.

All experiments were run for 30 more days after reaching a quasi-steady state, and the results presented here are time-averaged over these 30 days.

### Basal melt rates parameterization in the ice shelf-ocean model
Assuming the ice shelf acts as a perfect insulator between ocean and atmosphere, melting and freezing at the ice-ocean interface are parameterized using the widely applied three-equation formulation[45,77]. Basal melt rates $m_w$ are computed as $m_w = -\rho_{sw} c_w u_* \Gamma_T \Delta T / (\rho_{fw} L)$, with constant parameters: seawater density $\rho_{sw} = 1028 \, \text{kg m}^{-3}$, freshwater density $\rho_{fw} = 1000 \, \text{kg m}^{-3}$, specific heat capacity of seawater $c_w = 3974 \, \text{J} \, {}^\circ\text{C}^{-1} \, \text{kg}^{-1}$, heat transfer coefficient $\Gamma_T = 0.01$, and latent heat of fusion of ice $L = 3.34 \times 10^5 \, \text{J kg}^{-1}$. The thermal driving is given by $\Delta T = T_w - T_f$, where $T_w$ is the water temperature at the uppermost model layer, and $T_f = \lambda_1 S_b + \lambda_2 + \lambda_3 P_b$ is the local freezing point. The liquidus slope $\lambda_1 = -0.0573 \, {}^\circ\text{C}^{-1} \, \text{PSU}^{-1}$, intercept $\lambda_2 = 0.0832 \, {}^\circ\text{C}$, and pressure coefficient $\lambda_3 = -7.53 \times 10^{-8} \, {}^\circ\text{C} \, \text{Pa}^{-1}$ are constants. $S_b$ and $P_b$ are the salinity and pressure at the interface, respectively, with $P_b$ increasing with ice draft. The friction velocity $u_*$, representing the turbulent heat transfer efficiency, is given by $u_*^2 = C_D(u_w^2 + u_{res}^2)$, where $u_w$ is the ocean velocity at the uppermost model layer, $u_{res} = 1 \, \text{cm s}^{-1}$ represents subgrid-scale residual currents, and $C_D = 2.5 \times 10^{-3}$ is the drag coefficient at the ice-shelf base.

In summary, basal melt rates scale with the product of friction velocity and thermal driving, as $m_w \propto u_* \Delta T$. Accordingly, melt anomalies satisfy $\delta m_w \propto u_* \delta(\Delta T) + \Delta T \delta u_*$. Here, $\delta m_w$ is the melt rate anomaly, $\delta \Delta T$ is the thermal driving anomaly and $\delta u_*$ is the friction velocity anomaly that is proportional to the surface layer speed anomaly. It reflects how anomalies of two factors influence the melt rate anomalies: the influence of thermal-driving anomalies weighted by the local friction velocity and the influence of friction-velocity anomalies weighted by the local thermal driving. Given that the local thermal driving is in the order of $O(10^{-1})\,{}^\circ\text{C}$ and the local friction velocity is in the order of $O(10^{-3}) \, \text{m s}^{-1}$, as a result of their differing magnitudes, anomalies in friction velocity generally contribute more to melt rate variability than anomalies in thermal driving.

### Rossby number
The Rossby number $R_0$ is defined as the relative vorticity $\zeta = \frac{\partial v}{\partial x} - \frac{\partial u}{\partial y}$ normalized by the Coriolis parameter $f$ as $R_0 = \frac{\zeta}{f}$. Large absolute values of the Rossby number indicate the presence of energetic submesoscale eddies, which enhance the vertical advection of heat[41].

### Statistical analysis
Differences between ROUGH and SMOOTH draft experiments were assessed using a non-parametric Wilcoxon signed-rank test applied to paired grid-point basal melt rate anomalies (ROUGH relative to SMOOTH). Analyzes were conducted separately for the smoothing region and the deep-ice region under both COLD and WARM forcing. The test evaluates whether the median anomaly differs from zero and is therefore appropriate for identifying spatially consistent signals in the presence of non-Gaussian distributions and spatial heterogeneity. Statistical significance was evaluated using a two-sided test.

### Meltwater-mixing lines and reference salinity
Ocean-driven melting at the ice-shelf base results in mixing between glacial meltwater and the ambient source water. This produces a linear trajectory in temperature-salinity space known as the meltwater-mixing line, or Gade line[46]. It represents the linearized, small-amplitude limit of a more general enthalpy-conserving framework of meltwater-mixing relationships[78].

The slope of this line is given by $\frac{\partial T}{\partial S} = \frac{L}{S_0 c_w}$, with specific heat capacity of seawater $c_w = 4000 \, \text{J} \, {}^\circ\text{C}^{-1} \, \text{kg}^{-1}$, latent heat of fusion of ice $L = 3.34 \times 10^5 \, \text{J kg}^{-1}$ and salinity $S_0$ of the source water in contact with the ice. In the absence of other freshwater inputs, this relationship allows inversion for the source water mass by deriving the reference salinity, defined as the salinity at which the mixing line intersects the surface freezing point. This method has been applied to distinguish among High Salinity Shelf Water types beneath the Filchner−Ronne Ice Shelf[52]. In our study, reference salinities close to or below 34.2 indicate a CDW origin, while values closer to 35.35 suggest a WW origin.

### Fimbulisen ice dynamic model
The Fimbulisen ice dynamic model uses the Elmer/Ice dynamic ice sheet model[79] with the Shallow Shelf Approximation (SSA) to the Stokes equations[80]. The SSA assumes plug flow and hydrostatic balance, which give a very good approximation to the Stokes solution for the large scale horizontal flow in an ice shelf. The SSA also captures gravity-driven flow towards the apex of ice shelf channels, but with reduced accuracy due to deviations from plug flow. This discrepancy can be significant for the case of deeply incised channels in shallower sections of the shelf[24], but is lower and not well quantified for shallower channels in thick shelves such as our study area.

The model domain encompasses the central Jutulstraumen ice stream within Fimbulisen, using the same unstructured mesh as the corresponding portion of the Fimbulisen ice shelf-ocean model. Upper and lower surface elevations were derived from the REMA ice thickness data, assuming hydrostatic equilibrium with an ice density of 917kg m$^{-3}$ and a seawater density of 1028kg m$^{-3}$. Ice viscosity was determined using an inverse method to minimize differences between simulated and observed surface velocities[81]. Observed velocity fields were sourced from the MEaSUREs InSAR-Based Antarctica Ice Velocity Map (Version 2). The model assumes no basal melting, with surface mass balance set to a constant value based on this region's 1995–2014 temporal mean of the MAR dataset[82]. A transient simulation was conducted to estimate the ice shelf thinning rate.

The ice dynamic contribution to the thinning rate is given by the flux divergence,

$$\nabla \cdot (H\boldsymbol{u}) = H\nabla \cdot \boldsymbol{u} + u_x \frac{\partial H}{\partial x} + u_y \frac{\partial H}{\partial y}, \tag{1}$$

where $H$ is the ice thickness and $\boldsymbol{u} = (u_x, u_y)$ is the horizontal velocity vector. The first term represents thinning due to deformation, and the following terms reflect the effect of advection of thickness gradients. For our region of interest, the flux divergence signal is dominated by the advection of cross-flow features (keels and rifts) through the domain (Fig. S6). Given that our primary interest is in channel opening and closing rates, we therefore ignore the advection terms in our analysis and focus on deformational thinning. This gives the expected spatial pattern of gravity-driven channel closure, but with positive thinning values everywhere (Fig. S6b). These positive values are due to the fact that we have included along-flow extensional thinning (inherent in the first term on the right-hand side of Eq. (1)) but excluded large-scale advective thickening (inherent in the following terms), which would tend to counter the extensional thinning but without high spatial variability. Because our aim is to quantify how deformational thinning varies from outside the channel to inside the channel, we use a spatial anomaly rather than the absolute value. The spatial thinning rate anomaly presented in the main paper is thus calculated as the deviation of the deformational term from the mean value (3.17 m yr$^{-1}$) within a $10 \times 10$ km$^2$ area surrounding the example channel.

## Data availability

Input and output data for the Fimbulisen ice-shelf-ocean FVCOM simulations, together with input data for the Fimbulisen ice-dynamics Elmer/Ice simulations, are archived in the Norwegian NIRD Research Data Archive and are publicly available via a web-based interface (https://doi.org/10.11582/2026.o5mbgyn6). Post-processing MATLAB scripts used for figure generation are available from the corresponding author upon reasonable request.

## Code availability

The FVCOM source code and the Elmer/Ice source code used for this study are publicly available at https://doi.org/10.5281/zenodo.14570371 and https://doi.org/10.5281/zenodo.7892181, respectively.

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

## Acknowledgements

This research has been supported by the Research Council of Norway (RCN) through the KLIMAFORSK program through projects iMelt (project number 295075) and Clim2Ant (project number 343397). T.H. acknowledges funding from the RCN through its Centres of Excellence funding scheme via iC3, Centre for ice, Cryosphere, Carbon and Climate (project no. 332635) and from the European Union's Horizon 2020 programme (CRiceS, 101003826). C.Z. is the recipient of an Australian Research Council Discovery Early Career Researcher Award (project number DE240100267) funded by the Australian Government. C.Z. also received grant funding from the Australian Government as part of the Antarctic Science Collaboration Initiative program (ASCI000002) and the Australian-French Association for Research and Innovation and the Australian Research Council Special Research Initiative, Australian Centre for Excellence in Antarctic Science (project number SR200100008). R.G. was supported by the Research Council of Finland (project numbers 322430 and 355572), and by the Finnish Ministry of Education and Culture and CSC - IT Center for Science (Decision diary number OKM/10/524/2022). P.U. was supported by the Research Council of Finland (project number 355572). The authors wish to acknowledge Sigma2 HPC (project number nn9824k and nn11004k), Norway, and CSC - IT Center for Science, Finland, for providing computational resources.

## Author contributions

Q.Z. and T.H. contributed equally to this work. Together, they conceived the study, designed the experiments, conducted the formal analysis, and wrote the initial draft. Q.Z., C.Z., R.G., and T.H. implemented the simulations. A.M. provided the high-resolution ice draft topography, and J.L. contributed observational context. T.H., P.U., and R.G. acquired funding. Q.Z., T.H., R.G., C.Z., J.L, P.U., and A.M. contributed to the interpretation of the results and reviewed the manuscript.

## Funding

## Competing interests

The authors declare no competing interests.
