## [Transparent Peer Review file · Nature Communications]

Channelized topography amplifies melt-sensitivity of cold Antarctic ice shelves

Corresponding Author: Dr Qin Zhou

Version 0:

Reviewer comments:

Reviewer #1

(Remarks to the Author)

Zhou et al.'s "Channelized topography amplifies melt-sensitivity of cold Antarctic ice shelves" offers a significant contribution to our understanding of the impact of basal topography on ice-shelf cavity circulation and melt rates. I very much appreciated their experimental design, which entailed smoothing a portion of the ice shelf topography featuring channels and comparing circulation and melt rates between these two configurations (as well as between two ocean forcings). I also appreciate the use of an ice sheet model to compare melt rates with channel closure rates, though I had a question about the appropriateness of this model for that problem. Generally, I have noted places where I think the reader needs more clarification or discussion, but overall my comments constitute minor revisions. Where I believe more discussion is needed, I think the authors can achieve this in most or all instances with the addition of a sentence or two.

Major changes:

General comments:

I feel that there is a lack of discussion about how observations could be leveraged to validate your findings. For example, can satellite-derived melt rates be leveraged to determine melt rates in the Fimbulisen channels? Could repeat DEMs be used to detect lateral channel migration to the coriolis-favored side? How does the strength of CDW intrusions in your simulation compare with the observations that support CDW intrusions at Fimbulisen (your refs 32, 34-37) and is it plausible that the current levels of CDW intrusion are sufficient to maintain the observed channels through the mechanism you propose?

L69: I think the reader needs a little more information about the difference between COLD and WARM forcing here and L211 without having to refer to the cited papers to understand what SUM-30 means.

L122: The nature of the overturning in channels is unclear. I think what you're getting at is something like a spatially-confined overturning circulation pattern (mCDW-dominated?) superimposed above a large-scale overturning pattern (WW-dominated?).

L145: What exactly is the key ingredient in this chain?

Fig3: Did you mention somewhere in the text that there aren't significant differences in melt rate downstream of the deep region? I see that this is the case in Fig 3 but it's worth mentioning in the main text if it isn't there. I may have missed it.

Fig S6: Please add a panel showing the mean deformational pattern so that now the readers have visualizations for all terms if they also look at Fig 4a.4

L146: Wouldn't it also require a relatively quiescent grounding zone so that CDW and WW are not too thoroughly mixed? It seems likely that this would be the case but is perhaps worth pointing out to the reader. I.e., can you speak to how confident you are that the stratification between mCDW and WW would be preserved in reality?

L147: You have not yet discussed evidence or reasoning that supports that there is not significant subglacial discharge. Include here or the introduction.

L155: I might have missed something, but enhanced turbulent mixing didn't seem to be explicitly discussed earlier in the manuscript. To my reading, more emphasis was placed on changes in circulation patterns. I recommend either devoting more space to discussing the strength of turbulent mixing in your results or change the emphasis here to be consistent with the rest of the manuscript.

L211: if all simulations are initialized cold, how do you make the transition to warm conditions?

L237: There is a more accurate version of the Gade line in this paper <https://doi.org/10.1175/JPO-D-13-0253.1>. The differences are likely to be small but I recommend comparing the two lines (if you haven't already) prior to publication.

L245: I don't know enough about how ice sheet models would perform in the context of channel deformation, and I imagine that will also be true of many of your readers. Can you discuss the expected fidelity here, and whether the shallow shelf approximation would impact fidelity/accuracy?

L254: I'm a little surprised to see the thinning rate derived in a Eulerian frame. I would think that it would make a bit more sense to convert simulation results to a Lagrangian frame to more easily extract the deformational channel closure. I'm not convinced that the results are the same as simply excluding the advection terms from the equation you present (though maybe you can show that this is the case). The subtraction of the mean extensional thinning I can get on board with.

Fig 4b: it seems like this figure shows that in the channel the melt signal dominates the deformational signal under both cold and warm forcing, while outside the channel both terms are similar magnitudes. This point seems important to make in the text.

L148: Please elaborate on your reasoning here. I found this sentence hard to follow. Are you arguing that the channels in cold ice shelf cavities are a relict of past CDW intrusions and are currently closing through deformation? Or that CDW intrusions are period and the channels are sustained in a time mean sense?

L147: "primarily been attributed" this doesn't seem like a fair characterization of the literature to me. Lots of studies have pointed to melt-amplifying feedbacks. I think there should be more discussion of the literature on other ways channels can form.

L265: I do not believe the data or code availability statements yet comply with Nature Comms' policy.

Minor or technical changes:

L23: "and up to hundreds"

L28: "Likewise" the processes in this sentence and the previous one don't seem very similar

L30: "sustaining channel morphology" It would be helpful to have mentioned by this point that the key factor responsible for closing channels is ice deformation. Otherwise it's unclear why they need to be sustained.

L31: "deeper channel base" To me, the channel base could be confused with the channel crest. Perhaps "channel flanks" instead? Or do you mean the areas immediately outside the channel?

L70: "with COLD forcing" Specify which ice draft geometry was used here

L102: Present thermal driving anomalies in the same order here as above (positive then negative)

Fig 2 c,d Do you want to pick a different colorbar range so we can see some structure in (c)?

L120: is cooled >> "has cooled" to match "depleted"

L127: "This topographic trapping of the meltwater-driven overturning of the mCDW" hard to follow. Also, trapping and overturning sounds like a contradiction. Some other phrasing would be more clear.

L129: "surface flow" I think it's clearer to reserve this for surface currents outside the cavity. I think here you're talking about flow along the ice base

L131: because the keel is in the central part of the ice shelf as opposed to the shallower flank, right? Seems like a little bit of elaboration would be helpful here.

L134: "The comparison..." too much information is presented in this sentence. Break into several and elaborate.

L137-138: "spatial thinning anomaly... rather than absolute values" This contrast is unclear. Is it that the analysis approach is only capable of showing the former?

Fig S6a: Make explicit whether "total thinning rates" includes both ocean and ice as in Fig 4b or just ice results.

Fig S6b: Advection signal: is this the advection terms that you exclude from your thinning analysis? Make more explicit.

L148: "witness" odd word choice to me.

L235: what is FVCOM's EOS?

Fig S2 is this referenced in the text as a measure of robustness?

Fig S2: ROUGH?

Fig 4b: make ice flow line in this figure more prominent.

Carolyn Begeman

Reviewer #2

(Remarks to the Author)

Reviewer #3

(Remarks to the Author)

This paper investigates the impact of channelised basal topography and ocean conditions on both ice-shelf cavity circulation and melt rates on the Jutulstraumen ice stream. The paper presents interesting results which advance our knowledge of the role basal channels play in Antarctic ice shelf stability. Notably, the study focuses on a cold-water ice-shelf cavity, which unveils new circulation dynamics driven by basal topography. Furthermore, the study adds evidence that channelised topography could increase ice-shelf integrated melt rates.

In general, the quality of the manuscript and the results shown are consistent with publication in Nature Communications. However, the following comments should be addressed before the manuscript can be recommended for publication.

- In the abstract, the authors claim that basal channels 'focus thinning and reduce stability'. I believe that the overall role of channels on ice shelf stability is still a subject of debate, and so I would be careful making such a definitive statement.

- At the end of the abstract, the authors claim their results show that 'even modest CDW intrusions could destabilize cold Antarctic ice shelves'. The authors do show that the presence of basal channels increases the overall melt response to CDW intrusions from a 21% increase to a 35% increase. However, I don't think this is evidence that it could destabilise cold Antarctic ice shelves.

- The results presented show that introducing basal channels increases the overall ice-shelf integrated melt rate. As mentioned by the authors in the final paragraph lines 155-157), this opposes earlier idealised work. I think including this current state of knowledge in the introduction would help shape the reader's understanding of the significance of these results.

- In lines 30-31, the authors state that 'From an oceanic perspective, in the absence of these external factors, sustaining channelized morphology requires differential melting, ...'. If we are in the absence of all other factors, uniform melting would sustain a channel. If we are including ice's secondary flow, then the authors are right. Please clarify this statement.

- I am intrigued by the vertical resolution of the model used. If the model has 24 terrain following vertical layers, then the resolution depends on the water column thickness. By introducing a channel, you are increasing the local water column thickness and, therefore, presumably coarsening the vertical resolution. How do the authors think this will affect their results? Could this be contributing to the difference in melt seen between the SMOOTH and ROUGH ice base? It might be nice to see an attempt at making this more comparable between the different runs, perhaps by introducing more vertical layers in the ROUGH runs such that the average vertical resolution in the cavity is similar between the SMOOTH and ROUGH runs.

- The authors show that the increase in 'area integrated melt rates when changing from COLD to WARM forcing is four times larger under the ROUGH ice draft (0.4 m/yr) compared with the SMOOTH ice draft (0.1 m/yr)'. This is a really interesting result! However, the authors need to exert more care when using the term 'fourfold'. The use of the current statistic on line 153 is particularly overstated – 'In this way, even modest CDW incursions can trigger a fourfold increase in ice mass loss below deep ice...'. Firstly, as the study doesn't include the effects of melt on ice dynamics, I think 'mass loss' here should be replaced with 'melt rate'. This is the same in the abstract (line 16). While saying there has been a 'fourfold increase' is technically correct for the absolute values in the runs presented, making general claims using this statistic is misleading for the reader, as the baselines of the two cases are different. It would be beneficial if the authors made this clear within the text to avoid confusion. When discussing the result in a more general sense, it might worth considering using percentage change instead.

- While it is clear that the presence of channels leads to an increase in ice shelf integrated melt in this study, the argument would be strengthened if the authors could show that this increase is statistically significant. I suggest using a permutation test for matched pairs (for example, Wilcoxon's Signed Rank test) to compare the difference between the ROUGH and SMOOTH ice draft in both ocean conditions.

Figure 1:

- Please make the green dashed line more clear

- In e and f, please indicate which way round the channel is – is the reader looking forwards the grounding line or the calving front?

Figure 2:

- Again, please orient the transect used in c-f.

Figure 4:

- It's currently hard to tell the difference between the 'ice flow' line and the 'Ocean, COLD' line

Version 1:

Reviewer comments:

Reviewer #1

(Remarks to the Author)

I appreciate the authors' careful and thorough response to reviewers and related edits. This manuscript is a significant contribution to the field and I support its acceptance for publication.

Carolyn Begeman

Reviewer #2

(Remarks to the Author)

I would like to thank the authors for their thoughtful responses to my major comments. I noticed that only the major comments

were addressed, and the minor comments were missing. I apologise if my minor comments went missing from my initial assessment, but I would like to reiterate them here for the authors' consideration. I have updated them to only keep those pertinent after their first review. Overall, I think the authors have made significant improvements to the manuscript, and I would like to recommend for publication after minor revisions.

Minor Comments [Previous revision]

- [Line 86] It will be useful for the reader to add the percentage of the melt rate increase between simulations in a sentence here.
- [Line 96] I find this line confusing, consider rephrasing it. I would suggest saying: "In the deep-ice region (near the grounding line), the difference in the mean melt rate between the smooth warm and cold cases is small (0.1 m/yr), whereas in the rough warm and cold cases, the mean melt rate difference is 0.4 m/yr (4 times larger)."
- [Figure 2] Consider adjusting the colorbar in panel c, I am surprised that there is no salinity signature of melt water, despite the fact that the left flank of the channel is experiencing melting.
- [Line 103] Emphasise here the role of the velocity field and its resemblance to the melt rates. This is linked to my previous major comment.
- [Line 114] Add space between Figure and 2c.
- [Line 120] The analysis of the water mass properties is very interesting and an excellent addition to the manuscript.
- [Line 204] Add space between 1°C and at.
- [Line 228] Add space between Figure and 1d.

Additionally, I only found to minor typos in the new changes of the manuscript:

- [Line 159] Add space between observations and such.
- [Line 160] Add space between [49] and can.

Reviewer #3

(Remarks to the Author)

Overall, the authors have addressed the previous round of reviews exceptionally well. The paper will make a great contribution to the literature on our understanding of basal channels and their role in ice shelf stability in a cold cavity ice shelf. I think the paper is now in a position to be accepted for publication.

Channelized topography amplifies melt-sensitivity of cold Antarctic ice shelves

Summary

This study investigates the impact geometry has in the melt patterns in a cold cavity ice shelf (Fimbulisen Ice Shelf). The authors use four different simulations varying the ice shelf topography (smooth vs rough) and the ocean forcing (warm vs cold). The results show that in the warm case, rough topography leads to 4 times higher melting anomalies than in the smooth case. The authors attribute this to the intrusion of CDW into the cavity through the channels present in the rough topography case (mechanism absent in the smooth topography). Furthermore, the authors argue that cold cavities experiencing channelized melting are driven not only by subglacial discharge (previously thought to be the main driver), but also through past intrusions of CDW interacting with the ice shelf topography. This may have large impacts in the melt rates in the future as observations suggest an increase in the intrusions of CDW into the East Antarctic ice shelves. The study is a significant contribution to the field, since little attention has been put into exploring the sensitivity of cold ice shelves. The manuscript is well-written, and the figures are clear and informative, with careful attention paid to the congruency between the figures and the manuscript narrative. However, there are some areas that require clarification before I can recommend publication of the manuscript. In particular, the two major comments are described below. Below are my comments and suggestions for revisions.

Major Comments

Line 83-111 This section is a key stone of the manuscript, as it describes the impact of thermal driving in the melt patterns of the channel. However, I was surprised that little to no attention is given to the velocity (Coriolis favoured circulation) other than the mention in lines 96-99. Effectively, as the authors describe, both cold and warm cases experience velocities in the left flank of the channels up to 0.2 and 0.25 m/s, respectively. Yet, most of the discussion is centered around the thermal driving only, while in fact the conjunction of anomalous thermal drive and the velocity field results in the larger melt rates in the warm case. I recommend that the authors make this clearer within the manuscript. In fact, Gwyther et al., 2015 show that and I quote "Melting can be more dependent on the distribution of currents (u^*) rather than the distribution of thermal driving." This may be consistent with the fact that the melt rate anomalies shown in figure 2c-d seem to be better correlated with the velocity anomalies than the thermal drive anomaly. Furthermore, this may also explain why the melt rate anomalies in the base of the channels are negative in the warm case, despite the fact that the thermal driving is comparable to that experienced within the channel. Please also add a discussion on this.

Line 210 The author's setup is forced using the experiments SUM-30 and WIN-100. Looking into Hattermann et al., 2012. These experiments use different surface stress that, through Ekman transport, drive much more CDW into the cavity. Why are the authors not using analogous realisations using the same surface stress? That makes it tricky to compare the results between the two different cases. Therefore, a discussion or justification of this choice is needed, particularly because the simulations are forced with both the hydrographic and circulation of each realization.

Minor Comments

Abstract Where is “an order of magnitude” coming from? At best, the rough topography experiences 4 times higher melt rate anomalies. Please tone down or rephrase.

Figure 1 Consider changing the shading of the bars in the inset panel to make them more visible.

Line 82 It will be useful for the reader to add the percentage of the melt rate increase between simulations in a sentence here.

Line 89 This line is a bit confusing, consider rephrasing it. I would suggest saying: “In the deep-ice region (near the grounding line), the difference in the mean melt rate between the smooth warm and cold cases is small (0.1 m/yr), whereas in the rough warm and cold cases, the mean melt rate difference is 0.4 m/yr.”

Figure 2 Consider adding the melt rate anomaly as a line in panels e and f. It will make it easier to compare the anomalies between the different cases. Additionally, consider changing the colorbar in panel c, I am surprised that there is no salinity signature of melt water, despite the fact that the left flank of the channel is experiencing melting.

- Since figure 2 shows only the anomalies, I suggest adding a supplementary figure showing the absolute values of the melt rates, thermal driving, and velocities for the 4 cases (smooth warm/cold and rough warm/cold).

Line 94 Consider rephrasing this sentence to emphasise the role of the velocity field in addition to the thermal driving. See major comment.

Line 107 Add space between Figure and 2c.

Line 109 The thermal driving at the channel base is $sim + 0.15^{\circ}C$, not $sim + 0.25^{\circ}C$.

Line 106 & 111 Based on the colorbars of the figure, it seems to me that the difference between the melt rate anomalies of the rough warm and cold cases is around a factor of 2-3. Probably adding the melt rates within the figure will help to make this clearer. Make sure to reference the values within these sentences to make it clear to the reader.

Line 112 The analysis of the water mass properties is very interesting and an excellent addition to the manuscript.

Line 142 It seems to me that this is consistent with the preferential melting of channels on the Coriolis intensified flank, this probably should be acknowledged here.

Line 143 It is not clear to me that this is linked to basal channel formation, but rather it will affect the evolution of the basal channel, particularly because the melt rate anomalies are not centered within the channel crest but on the Coriolis-favored flank. Please rephrase. However, I agree that the coexistence of water masses impacts the melt rates within the basal channels.

Figure 4 Consider making the dashed lines more visible (e.g., thicker).

Line 157 I recognise the novelty of this study in the context of describing the impact of channelized topography in cold cavity ice shelves. However, I think the authors need to tone down the statement “the novel channelized overturning mechanism”. It is known that CDW intrusions are the key mechanism driving melting of the ice shelves (Rosevear et al., 2025) and that basal channels tend to channelize properties (Cheng et al., 2024).

Line 178 Add space between $1^{\circ}C$ and at.

Line 187 What is the vertical resolution within the channel section of Figure 2? It will be useful to add this information to assess the representation of the boundary layer.

Line 208 Consider adding more information on the model setup to help the reader understand the model. Also, at what frequency are the boundaries applied? See second major comment.

Line 201 Add space between Figure and 1d.

- Neither Extended Data Figure 5 nor 6 is referenced in the text. Please add references to them or remove them.
- Data availability statement only says that the simulations will be publicly available. Please add more information on where the data will be available, i.e., the mentioned "web-based interface".

References

1. Gwyther, D. E., Galton-Fenzi, B. K., Dinniman, M. S., Roberts, J. L., & Hunter, J. R. (2015). The effect of basal friction on melting and freezing in ice shelf-ocean models. *Ocean Modelling*, 95, 38-52.
2. Rosevear, M. G., Gayen, B., Vreugdenhil, C. A. & Galton-Fenzi, B. K. How Does the Ocean Melt Antarctic Ice Shelves? *Annu. Rev. Mar. Sci.* 17, 325-353 (2025).
3. Cheng, C. et al. Ice shelf basal channel shape determines channelized ice-ocean interactions. *Nat. Commun.* 15, 2877 (2024).

Reviewer #1

Zhou et al.’s “Channelized topography amplifies melt-sensitivity of cold Antarctic ice shelves” offers a significant contribution to our understanding of the impact of basal topography on ice-shelf cavity circulation and melt rates. I very much appreciated their experimental design, which entailed smoothing a portion of the ice shelf topography featuring channels and comparing circulation and melt rates between these two configurations (as well as between two ocean forcings). I also appreciate the use of an ice sheet model to compare melt rates with channel closure rates, though I had a question about the appropriateness of this model for that problem. Generally, I have noted places where I think the reader needs more clarification or discussion, but overall my comments constitute minor revisions. Where I believe more discussion is needed, I think the authors can achieve this in most or all instances with the addition of a sentence or two.

We thank the reviewer for her time and effort reviewing our manuscript and for her thoughtful and supportive feedback and useful suggestions that helped us to improve our manuscript. Below, we respond to each comment in turn, with our responses indicated in blue.

Major changes:**General comments:**

I feel that there is a lack of discussion about how observations could be leveraged to validate your findings. For example, can satellite-derived melt rates be leveraged to determine melt rates in the Fimbulisen channels? Could repeat DEMs be used to detect lateral channel migration to the coriolis-favored side? How does the strength of CDW intrusions in your simulation compare with the observations that support CDW intrusions at Fimbulisen (your refs 32, 34-37) and is it plausible that the current levels of CDW intrusion are sufficient to maintain the observed channels through the mechanism you propose?

We agree with the reviewer that a discussion of how observations could be leveraged to validate our findings would strengthen the manuscript. In the Implications for ice shelf stability section, we have therefore added the following paragraph (Lines 155–161 in the revised manuscript) discussing observational constraints on differential melting and the evolution of basal channels:

"Recent advances in spatially refined satellite-derived estimates of basal melt rates may help identify patterns of differential melting [47]. However, uncertainties of several metres per year remain large relative to the expected signals beneath moderately melting ice shelves such as Fimbulisen [35, 40]. Repeated Digital Elevation Models may reveal channel migration and evolution[48], but this approach requires disentangling ice-advection effects from melt-driven thickness changes at individual channels on interannual timescales. Ultimately, ground-based, process-oriented observations, such as time series from autonomous phase-

sensitive radio-echo sounders[49], can provide accurate reference measurements, but require dedicated field deployments and resources."

In addition, we have added the following paragraph (Lines 171–175 of the revised manuscript) to relate the simulated strength of CDW intrusions to available observations beneath Fimbulisen:

"The time-averaged forcing with COLD and WARM conditions in our experiments is chosen to clearly delineate the effect of the CDW intrusions that are observed to occur in pulses below Fimbulisen [35]. Despite this idealization, the magnitude and vertical extent of the near-bottom temperature increase under the WARM forcing are consistent with ice-shelf borehole observations [37,52]. If such temperature anomalies propagate farther into the cavity, as inferred by [53], episodic warm inflows may therefore contribute to maintaining the channelized geometry observed at the Jutulstraumen ice stream."

Specific Comments

L69: I think the reader needs a little more information about the difference between COLD and WARM forcing here and L211 without having to refer to the cited papers to understand what SUM-30 means.

We thank the reviewer for this suggestion. To make the distinction between the COLD and WARM forcing self-contained, we have revised the Methods to explicitly explain the differences between them. We also explicitly explain the physical meaning of the ROMS experiment names, so that the interpretation of COLD and WARM forcing does not require consultation of the cited ROMS studies.

Specifically, Lines 208-212 of the original manuscript have been replaced by the following sentences (Lines 235-245 of the revised version):

"Lateral boundary conditions were taken from the eddy-resolving Fimbulisen ice shelf–ocean model based on the Regional Ocean Model System (ROMS) [34], which has been used to investigate basal melting and cavity circulation beneath Fimbulisen under different CDW inflow scenarios that have been observed [35]. The variability of CDW inflow is controlled by the wind stress and hydrographic forcing, with experiment names reflecting combinations of winter or summer hydrography and weak (30), mean (100), and strong (130) wind forcing. From the 18 scenarios, we selected the WIN-100 and SUM-30 climatologies as lateral boundary conditions for the COLD and WARM forcing experiments, respectively. WIN-100 represents a scenario with a homogeneous WW-filled cavity and no CDW inflow, and SUM-30 represents a scenario with constant CDW inflow to the cavity, resulting the highest melt rates in the deep-ice region. All simulations were initialized from the WIN-100 climatology. In the COLD forcing experiments, cavity conditions consistent with WIN-100 hydrography and

circulation were maintained through the lateral open-boundary conditions. In the WARM experiments, the cavity transitions from the initial COLD state to WARM conditions through lateral open-boundary forcing derived from the SUM-30 climatology."

L122: The nature of the overturning in channels is unclear. I think what you're getting at is something like a spatially-confined overturning circulation pattern (mCDW-dominated?) superimposed above a large-scale overturning pattern (WW-dominated?).

We thank the reviewer for this insightful comment. Our simulations indeed exhibit two distinct but related overturning circulations. The first is a cavity-scale overturning associated with the classical melt–buoyancy pump, which is present in all experiments. Superimposed on this in the WARM forcing experiments is a spatially confined, density-driven overturning that arises from meltwater-driven density changes of two distinct water masses (CDW and WW) in the cavity. This localized, density-driven overturning occurs near the grounding line whenever CDW is present. Only in the ROUGH draft experiments, however, is the overturned, meltwater-modified CDW trapped within basal channels, where it enhances local melting and the associated buoyant flux. This enhanced buoyancy flux, in turn, strengthens the cavity-scale overturning.

To clarify this distinction in the manuscript, we have explicitly added the term "density-driven" to describe the localized overturning in Lines 131 and 168 of the revised manuscript.

L145: What exactly is the key ingredient in this chain?

The key ingredient is the coexistence of distinct water masses beneath the ice shelf, arranged such that a warmer and initially denser water mass (CDW) underlies a colder and lighter water mass (WW). When these water masses interact with the ice base, their properties evolve along separate meltwater-mixing lines. As a result, the originally denser CDW can become sufficiently buoyant through mixing with meltwater while remaining warmer than the overlying WW. This density reversal generates a localized, density-driven overturning circulation.

To clarify this, we have now revised Lines 145-146 of the original manuscript to the following sentences (Lines 166-169 of the revised version), as:

"The key ingredient in this mechanism is the coexistence of distinct water masses, with warmer, initially denser CDW underlying colder, lighter WW. Because their properties evolve along separate meltwater-mixing lines, CDW can become buoyant while remaining warm, generating a localized density-driven overturning that causes warmer water to be trapped inside the channels."

Fig3: Did you mention somewhere in the text that there aren't significant dif-

ferences in melt rate downstream of the deep region? I see that this is the case in Fig 3 but it's worth mentioning in the main text if it isn't there. I may have missed it.

We did not explicitly state in the original manuscript that melt-rate differences between the ROUGH and SMOOTH draft experiments are small downstream of the deep-ice region. We have now clarified this point in the Results section by explicitly noting that basal channels imprint both the spatial distribution of melting and melting difference within the deep-ice region, and the spatial pattern of melt-rate differences within the deep-ice region, whereas downstream of this region the melt-rate differences are small. Accordingly, we have added to the revised manuscript the following phrase (Lines 81-82) :

"and on the spatial pattern of melt rate differences between the ROUGH and SMOOTH draft experiments"

and the following phrase (Lines 85-86):

"Downstream of this region, melt-rate differences are small due to the diminishing ice draft difference (Extended Data Figure 1c)."

Fig S6: Please add a panel showing the mean deformational pattern so that now the readers have visualizations for all terms if they also look at Fig 4a.4

We thank the reviewer for this suggestion. We have added a panel showing the mean deformational thinning pattern to Extended Data Figure 6 (panel b), so that all terms contributing to the total thinning rate are explicitly visualized alongside the total and advection-induced components, consistent with Figure 4a.

L146: Wouldn't it also require a relatively quiescent grounding zone so that CDW and WW are not too thoroughly mixed? It seems likely that this would be the case but is perhaps worth pointing out to the reader. I.e., can you speak to how confident you are that the stratification between mCDW and WW would be preserved in reality?

We thank the reviewer for this insightful comment. The mechanism discussed here is not limited to a quiescent grounding-line zone; it applies wherever meltwater-modified CDW becomes less dense than the overlying WW. At Fimbulisen, CDW enters at depth and flow along the sea floor into the cavity, remains overlain by more bouyant and colder WW till the grounding-zone vicinity. There, interaction with the ice base produces meltwater-modified CDW that becomes locally buoyant while still warmer than the overlying WW, driving localized overturning circulation and confinement of warm water within basal channels.

Although CDW is progressively modified by mixing with WW along its pathway, the stratification between CDW and WW is preserved at the cavity scale, as indicated qualitatively by the -1.7°C isothermal line in Figure 1d. We have clarified this point in the revised manuscript (Line 124) by adding the following phrase:

"while the stratification with WW is preserved (Figure 1d),"

L147: You have not yet discussed evidence or reasoning that supports that there is not significant subglacial discharge. Include here or the introduction.

We thank the reviewer for raising this point. We do not assume that subglacial discharge is absent in the Jutulstraumen region; rather, we acknowledge that subglacial discharge may occur and has previously been linked to channelized melting beneath colder ice shelves. The key point of this study is that our simulations demonstrate that channelized basal morphology can be sustained by oceanic processes alone, independent of subglacial discharge. Any contribution from subglacial discharge would therefore act in addition to, and potentially amplify, the mechanism identified here. We have clarified this distinction in the revised manuscript by removing Lines 147-148 of the original version and adding the following sentences (Lines 162-166 of the revised version):

"Channelized melting beneath colder ice shelves such as Fimbulisen has previously often been linked to subglacial discharge [19, 50] and we assume that similar processes are also at play at the Jutulstraument grounding line [51]. However, the simulations presented in this study provide, to our knowledge, the first explicit example of an ocean-driven process that may promote basal channel growth beneath a slowly melting ice shelf through differential melting, which has previously been attributed to warm water cavities [17]."

L155: I might have missed something, but enhanced turbulent mixing didn't seem to be explicitly discussed earlier in the manuscript. To my reading, more emphasis was placed on changes in circulation patterns. I recommend either devoting more space to discussing the strength of turbulent mixing in your results or change the emphasis here to be consistent with the rest of the manuscript.

We agree with the reviewer's comment. To ensure consistency with the overall emphasis of the manuscript, we have revised Lines 155–156 of the original manuscript to de-emphasize turbulent mixing and instead focus on circulation-driven effects associated with small-scale basal topography. Specifically, we replaced these lines with the following sentence (Lines 184–185 of the revised version):

"Overall, our findings show small-scale basal topography can enhance the area average basal melting by promoting vertical heat transfer to the ice base [41], rather than stabilizing ice shelves as was suggested by earlier idealized mod-

elling[12,13]. "

L211: if all simulations are initialized cold, how do you make the transition to warm conditions?

In the WARM forcing experiments, the transition from the initial COLD state to WARM conditions is achieved by imposing lateral open-boundary forcing derived from the SUM-30 climatology. As the model integrates forward in time, the boundary-forced warm water progressively advects into the cavity and replaces the initial cold conditions. To clarify this point, we have added the following sentences to the revised manuscript (Lines 243–245):

"In the COLD forcing experiments, cavity conditions consistent with WIN-100 hydrography and circulation were maintained through the lateral open-boundary conditions. In the WARM experiments, the cavity transitions from the initial COLD state to WARM conditions through lateral open-boundary forcing derived from the SUM-30 climatology."

L237: There is a more accurate version of the Gade line in this paper <https://doi.org/10.1175/JPO-D-13-0253.1>. The differences are likely to be small but I recommend comparing the two lines (if you haven't already) prior to publication.

We thank the reviewer for bringing this paper to our attention. In this study, meltwater-mixing relationships are diagnosed using potential temperature and salinity, following the classical Gade formulation. This diagnostic approach is consistent with the model representation of ice–ocean interaction, which is prescribed through a melt parameterization and associated virtual salt flux based on the standard Gade line. McDougall et al. (2014) presents a more general, enthalpy-conserving framework for meltwater mixing, of which the Gade line represents the linearized, small-amplitude limit. We have therefore acknowledged this more general framework by adding the corresponding citation in Lines 286–287 of the revised manuscript:

"It represents the linearized, small-amplitude limit of a more general enthalpy-conserving framework of meltwater-mixing relationships [78]."

L245: I don't know enough about how ice sheet models would perform in the context of channel deformation, and I imagine that will also be true of many of your readers. Can you discuss the expected fidelity here, and whether the shallow shelf approximation would impact fidelity/accuracy?

We thank the reviewer for raising this point. Quantifying the impact of using the shallow-shelf approximation (SSA) versus a full-Stokes formulation on channel deformation is indeed challenging. We have therefore added a paragraph discussing the expected fidelity of the ice-dynamic model in this context and the potential implications of the SSA assumption, together with a relevant

reference. We note that the impact of the SSA versus Stokes choice is likely larger in the cited study, owing to their shallower ice shelf and deeper channels compared to Fimbulisen, which makes a direct quantitative comparison difficult. Assessing the sensitivity of channel closure rates to the SSA assumption using full-Stokes simulations is an important direction for future work. Specifically, the following sentences have been added to the Methods section in the revised manuscript (Lines 296-300):

" (SSA) to the Stokes equations [80]. The SSA assumes plug flow and hydrostatic balance, which give a very good approximation to the Stokes solution for the large scale horizontal flow in an ice shelf. The SSA also captures gravity-driven flow towards the apex of ice shelf channels, but with reduced accuracy due to deviations from plug flow. This discrepancy can be significant for the case of deeply incised channels in shallower sections of the shelf [24], but is lower and not well quantified for shallower channels in thick shelves such as our study area."

L254: I'm a little surprised to see the thinning rate derived in a Eulerian frame. I would think that it would make a bit more sense to convert simulation results to a Lagrangian frame to more easily extract the deformational channel closure. I'm not convinced that the results are the same as simply excluding the advection terms from the equation you present (though maybe you can show that this is the case). The subtraction of the mean extensional thinning I can get on board with.

We thank the reviewer for this thoughtful comment. Our aim here is to isolate the deformational component of the flux divergence, which has been shown to be the dominant contributor to channel filling (Wearing et al., 2021). This component primarily reflects secondary, cross-flow deformation rather than advection associated with the along-flow motion, and therefore is expected to differ only weakly between an Eulerian and a Lagrangian reference frame.

Strictly speaking, we agree that the net thickness change diagnosed in a Lagrangian frame is not mathematically identical to the deformational thinning rate obtained by excluding advection terms. However, given that the deformation is dominated by cross-flow processes, we expect the resulting difference to be small compared to other sources of model uncertainty.

We also note that the ice-dynamic simulations in this study are performed on a fixed, Eulerian mesh. While it would in principle be possible to remap model output to a Lagrangian framework for diagnostic purposes, such a transformation would itself introduce additional numerical uncertainty. As the reviewer does not explicitly request changes to the manuscript text, and because we do not expect the choice of reference frame to alter our conclusions, we have not modified the main text at this stage. We would be happy to revise the manuscript further at the editor's request.

Fig 4b: it seems like this figure shows that in the channel the melt signal dominates the deformational signal under both cold and warm forcing, while outside the channel both terms are similar magnitudes. This point seems important to make in the text.

We thank the reviewer for this comment. It is important to note that the deformational thinning rate shown here is expressed as a spatial anomaly rather than an absolute value. Accordingly, the meaningful comparison is not the absolute magnitude of deformational thinning versus melt rates, but how the deformational thinning varies from outside the channel to inside the channel.

When viewed in this way, the spatial variation in deformational thinning is comparable to the variation in basal melt rates under COLD forcing, whereas under WARM forcing the melt rates exhibit a substantially larger contrast between the channel interior and the surrounding ice. This distinction is reflected in Figure 4b, where the term “thinning rate anomaly” is explicitly used in both the color bar and the figure caption.

To further clarify this point and our reasoning, we have added the following sentence to the revised manuscript (Lines 318–319), at the end of the ice-dynamic model description where the deformational calculation is introduced:

"Because our aim is to quantify how deformational thinning varies from outside the channel to inside the channel, we use a spatial anomaly rather than the absolute value."

L148: Please elaborate on your reasoning here. I found this sentence hard to follow. Are you arguing that the channels in cold ice shelf cavities are a relict of past CDW intrusions and are currently closing through deformation? Or that CDW intrusions are period and the channels are sustained in a time mean sense?

We agree with the reviewer that this statement was ambiguous. Our intention was to highlight that the presence of channels beneath cold water ice shelves in places where subglacial outflows are absent may be a proxy for (present or past) CDW inflows. When addressing an earlier reviewer request on improving the discussion on the role of observations, the sentence has been removed.

L147: "primarily been attributed" this doesn't seem like a fair characterization of the literature to me. Lots of studies have pointed to melt-amplifying feedbacks. I think there should be more discussion of the literature on other ways channels can form.

We agree with the reviewer that the broader literature on basal channel formation encompasses a range of melt-amplifying feedbacks, particularly beneath warm ice shelves influenced by relatively warm cavity waters. These processes

are referenced in the revised Introduction (Lines 32–33), where we discuss the variety of mechanisms that have been proposed to contribute to basal channel formation.

Our original wording was intended to refer specifically to cold ice-shelf cavities, where regional studies have most commonly invoked subglacial discharge as a driver of channelized melting. However, we recognize that this distinction was not sufficiently clear in the original phrasing. In response to an earlier reviewer request to improve the discussion of subglacial discharge, we have therefore removed the sentence in question from the manuscript.

L265: I do not believe the data or code availability statements yet comply with Nature Comms' policy.

Thanks for the comment. We have modified the data availability statements in the revised manuscript (Lines 323-326), as

"Input and output data for the Fimbulisen ice-shelf–ocean FVCOM simulations, together with input data for the Fimbulisen ice-dynamics Elmer/Ice simulations, are archived in the Norwegian NIRD Research Data Archive and are publicly available via a web-based interface (<https://doi.org/10.11582/2026.o5mbgyn6>). Post-processing MATLAB scripts are available from the corresponding author upon reasonable request."

Minor or technical changes:

L23: "and up to hundreds"

Corrected.

L28: "Likewise" the processes in this sentence and the previous one don't seem very similar

"Likewise" has been replaced by "In addition".

L30: "sustaining channel morphology" It would be helpful to have mentioned by this point that the key factor responsible for closing channels is ice deformation. Otherwise it's unclear why they need to be sustained.

Agree. We have added the phrase "against ice creep closure [23, 24 ,25]." to the revised version of the manuscript (Line 34).

L31: "deeper channel base" To me, the channel base could be confused with the channel crest. Perhaps "channel flanks" instead? Or do you mean the areas immediately outside the channel?

We agree with the reviewer that the term “deeper channel base” could be confusing. In this context, “channel base” refers to the deepest part of the channel, whereas the channel crest corresponds to the shallowest part. To avoid ambiguity, we have clarified this terminology by explicitly defining the channel base as “the deepest part of the channel” when the term is first introduced in the revised manuscript (Line 36).

L70: "with COLD forcing" Specify which ice draft geometry was used here

Specified.

L102: Present thermal driving anomalies in the same order here as above (positive then negative)

We thank the reviewer for this comment. In this sentence, thermal-driving anomalies are described following the spatial progression along the channel cross-section (from the channel base toward the crest, corresponding to left to right in Figure 2e,f), rather than being ordered by sign. We consider this ordering be consistent with the figure and therefore have not modified the text.

We note, however, that we identified and corrected an error in the description of thermal-driving anomalies under WARM forcing. The original phrase *"causing positive thermal driving anomalies (relative to the SMOOTH topography) that range from $\sim -0.05^\circ\text{C}$ at the channel base to $\sim +0.25^\circ\text{C}$ near the channel crest"* has been revised to *"causing positive thermal driving anomalies (relative to the SMOOTH topography) of $\sim +0.15^\circ\text{C}$ from the channel base until 0.3 km upstream of the apex."* in the revised manuscript (Lines 115-116).

Fig 2 c,d Do you want to pick a different colorbar range so we can see some structure in (c)?

We thank the reviewer for this suggestion. Figure 2c shows the reference salinity, which by definition reflects the source water mass modified by meltwater mixing along the Gade line. Under COLD forcing, the water within the channel is dominated by WW and therefore exhibits a nearly uniform reference salinity (~ 34.35). As a result, adjusting the colorbar range would not be expected to reveal additional spatial structure in this panel. We note, however, that temperature variations within the channel are present and are shown in Figure 1e.

L120: is cooled » "has cooled" to match "depleted"

Corrected.

L127: "This topographic trapping of the meltwater-driven overturning of the mCDW" hard to follow. Also, trapping and overturning sounds like a contradiction. Some other phrasing would be more clear.

We agree that the original phrasing was unclear. The overturning mechanism itself is described in the preceding paragraph; in this sentence, we are referring to the subsequent confinement of the already overturned, meltwater-modified CDW within basal channels. To clarify the sequence of processes and avoid any perceived contradiction between overturning and trapping, we have revised the original phrase to "*This topographic confinement of the overturned meltwater-modified mCDW*" in the revised manuscript (Line 136).

"Surface flow" I think it's clearer to reserve this for surface currents outside the cavity. I think here you're talking about flow along the ice base.

Agree. The term "surface flow" has been replaced with "flow along the ice base" in the revised manuscript (Line 139).

L131: because the keel is in the central part of the ice shelf as opposed to the shallower flank, right? Seems like a little bit of elaboration would be helpful here.

We agree that additional clarification is helpful here. The absence of meltwater-modified mCDW along the Jutulstraumen keel under the SMOOTH draft is not related to the keel's central position relative to the flanks, but rather to the removal of basal channels. Smoothing eliminates the channelized topography and the associated topographically guided along-channel flow, preventing the buoyant, meltwater-modified mCDW from being confined near the keel and instead allowing it to be flushed toward the shallower flanks. To clarify this mechanism, we have added the following sentences to the revised manuscript (Lines 138-139):

"smoothing removes the basal channels along the keel and thus the topographically guided along-channel flow. Consequently, "

L134: "The comparison... " too much information is presented in this sentence. Break into several and elaborate.

Following the reviewer's comment, we have revised the original long sentence by splitting it into several shorter sentences and providing additional elaboration (Lines 144–148 of the revised manuscript):

"Enhanced melting within the basal channels, driven by the trapped meltwater-modified CDW, results in differential basal melting across the channel. To assess the impact of this differential melting on the channel evolution, we compare basal melt rates across the example channel (Figure 4) with ice-dynamic thinning rates derived from a stand-alone ice-dynamic model(Methods). This comparison suggests an instrumental role of the occasional inflows of CDW below Fimbulisen in maintaining the observed channelized geometry near the Jutulstraumen keel.
"

L137-138: "spatial thinning anomaly... rather than absolute values" This contrast is unclear. Is it that the analysis approach is only capable of showing the former?

We thank the reviewer for this comment. The use of a spatial thinning-rate anomaly is a deliberate analysis choice rather than a limitation of the method. Because our primary interest is in the relative contrast between deformational thinning inside and outside basal channels, we focus on deformational thinning after excluding the large-scale advection terms. This yields positive thinning values everywhere, dominated by along-flow extensional thinning, which obscures the channel-scale contrast when expressed in absolute terms.

To isolate the physically relevant variation across the channel, we therefore analyze spatial anomalies relative to the local mean. This rationale is explained in detail in the Methods section (Lines 311–319 of the revised manuscript).

Fig S6a: Make explicit whether "total thinning rates" includes both ocean and ice as in Fig 4b or just ice results.

We have revised the caption of Extended Data Figure 6 in the revised manuscript to explicitly state that the thinning rates shown are derived from the Fimbulisen ice-dynamic model.

Fig S6b: Advection signal: is this the advection terms that you exclude from your thinning analysis? Make more explicit.

We have clarified in the updated caption that the advection term is shown for completeness but is not used in the thinning-rate analysis; only deformation-induced thinning rates are compared with the ocean-driven melt rates.

L148: "witness" odd word choice to me.

The sentence including this word have been removed in response to an earlier reviewer request to improve the discussion of subglacial discharge.

L235: what is FVCOM's EOS?

FVCOM supports several equations of state. In all simulations presented here, in-situ density is computed using the polynomial equation of state (Jackett and McDougall, 1995), which accounts for pressure effects. We have added this information to the Methods section for clarity (Lines 209–210 of the revised manuscript).

Fig S2 is this referenced in the text as a measure of robustness?

Yes. Figure S2 is referenced in the text as a robustness check (Line 71 of the original manuscript and Line 75 of the revised manuscript).

Fig S2: ROUGH?

Yes. We have clarified this in the updated caption by explicitly stating that Figure S2 corresponds to the ROUGH draft configuration.

Fig 4b: make ice flow line in this figure more prominent.

We have increased the line thickness of the "ice flow" line (blue dashed line) in Figure 4 to improve its visibility.

Reviewer #2

Summary

This study investigates the impact geometry has in the melt patterns in a cold cavity ice shelf (Fimbulisen Ice Shelf). The authors use four different simulations varying the ice shelf topography (smooth vs rough) and the ocean forcing (warm vs cold). The results show that in the warm case, rough topography leads to 4 times higher melting anomalies than in the smooth case. The authors attribute this to the intrusion of CDW into the cavity through the channels present in the rough topography case (mechanism absent in the smooth topography). Furthermore, the authors argue that cold cavities experiencing channelized melting are driven not only by subglacial discharge (previously thought to be the main driver), but also through past intrusions of CDW interacting with the ice shelf topography. This may have large impacts in the melt rates in the future as observations suggest an increase in the intrusions of CDW into the East Antarctic ice shelves. The study is a significant contribution to the field, since little attention has been put into exploring the sensitivity of cold ice shelves. The manuscript is well-written, and the figures are clear and informative, with careful attention paid to the congruency between the figures and the manuscript narrative. However, there are some areas that require clarification before I can recommend publication of the manuscript. In particular, the two major comments are described below. Below are my comments and suggestions for revisions.

We thank the reviewer for the thoughtful and supportive feedback and useful suggestions that helped us to improve our manuscript. Below, we respond to each comment in turn, with our responses indicated in blue. Owing to its length, the first comment has been divided into two parts.

Major Comments

*Line 83-111: **Part 1)** This section is a key stone of the manuscript, as it describes the impact of thermal driving in the melt patterns of the channel. However, I was surprised that little to no attention is given to the velocity (Coriolis favoured circulation) other than the mention in lines 96-99. Effectively, as the authors describe, both cold and warm cases experience velocities in the left flank of the channels up to 0.2 and 0.25 m/s, respectively. Yet, most of the discussion is centered around the thermal driving only, while in fact the conjunction of anomalous thermal drive and the velocity field results in the larger melt rates in the warm case. I recommend that the authors make this clearer within the manuscript.*

We agree with the reviewer that the role of friction velocity deserves more explicit attention when describing the impact of basal channels on melting under both COLD and WARM forcing. In response, we have revised the description to better emphasize the combined influence of friction velocity and thermal driving on basal melt rates in the warm case. Specifically, we have revised Lines 109–110 of the original manuscript to the following sentences in the revised ver-

sion (Lines 116–118):

"While the topographically enhanced friction velocity generally increases melting under the ROUGH ice draft, this alternate thermal structure under WARM forcing additionally amplifies the effect of the enhanced ocean speed inside the channel (Figure 2f)."

Line 83-111: Part 2) In fact, Gwyther et al., 2015 show that and I quote "Melting can be more dependent on the distribution of currents (u^) rather than the distribution of thermal driving." This may be consistent with the fact that the melt rates anomalies shown in figure 2c-d seem to be better correlated with the velocities anomalies than the thermal drive anomaly. Furthermore, this may also explain why the melt rate anomalies in the base of the channels are negative in the warm case, despite the fact that the thermal driving is comparable to that experienced within the channel. Please also add a discussion on this.*

We agree with the reviewer that "Melting can be more dependent on the distribution of currents (u^*) rather than the distribution of thermal driving". To clarify this point, we have added the following paragraph in the Methods section of the revised manuscript (Lines 267-273) where basal melt rates parameterization is presented:

"In summary, basal melt rates scale with the product of friction velocity and thermal driving, as $m_w \propto u_ \Delta T$. Accordingly, melt anomalies satisfy $\delta m_w \propto u_* \delta(\Delta T) + \Delta T \delta u_*$. Here, δm_w is the melt rate anomaly, $\delta \Delta T$ is the thermal driving anomaly and δu_* is the friction velocity anomaly that is proportional to the surface layer speed anomaly. It reflects how anomalies of two factors influence the melt rate anomalies: the influence of thermal-driving anomalies weighted by the local friction velocity and the influence of friction-velocity anomalies weighted by the local thermal driving. Given that the local thermal driving is in the order of $O(10^{-1})$ °C and the local friction velocity is in the order of $O(10^{-3})$ ms^{-1} , as a result of their differing magnitudes, anomalies in friction velocity generally contribute more to melt rate variability than anomalies in thermal driving."*

This framework explains why melt-rate anomalies in Figure 2c–d correlate more strongly with velocity anomalies than with thermal-driving anomalies, and also why melt-rate anomalies at the channel base can be negative under WARM forcing despite comparable thermal driving within the channel.

Line 210 The author’s setup is forced using the experiments SUM-30 and WIN-100. Looking into Hattermann et al., 2012. These experiments use different surface stress that, through ekman transport, drive much more CDW into the cavity. Why are the authors not using analogous realisations using the same surface stress? That makes it tricky to compare the results between the two different cases. Therefore, a discussion or justification of this choice is needed, particularly because the simulations are forced with both the hydrographic and circulation of each realization.

We thank the reviewer for raising this important point. In the ROMS experiments of Hattermann et al. (2014), variability in modified Circumpolar Deep Water (CDW) inflow is indeed controlled by the combined effects of surface stress forcing and hydrographic conditions, which together regulate the depth and strength of CDW access to the cavity.

In the present study, however, the FVCOM simulations are not forced directly by surface stress. Instead, the influence of surface wind forcing is incorporated implicitly through the lateral open-boundary conditions, using prescribed hydrographic and circulation climatologies derived from selected ROMS realizations. In this framework, the depth and presence of CDW at the open boundary encapsulate the net effect of surface wind stress and hydrographic forcing in the parent ROMS simulations.

From the full ensemble of 18 ROMS realizations spanning a range of mCDW inflow strengths, we selected WIN-100 and SUM-30 as representative end-member states for the COLD and WARM cavity configurations, respectively. WIN-100 corresponds to a regime characterized by a cavity filled with cold Winter Water and no CDW inflow, whereas SUM-30 represents a state with sustained CDW intrusion reaching the deep ice base and producing the strongest deep-ice melt rates.

Our objective is therefore not to compare the detailed surface wind stress forcing between WIN-100 and SUM-30, but rather to contrast two physically distinct cavity states—one without and one with sustained CDW access—in order to isolate how basal channels modulate ice–ocean interactions under COLD versus WARM cavity conditions. Because both the hydrographic structure and the associated circulation are prescribed consistently at the lateral boundary for each case, the comparison remains internally consistent within the scope of our modeling approach.

To clarify this point, and in response to Reviewer 1, we have refined the description of the forcing in the Methods section in the revised manuscript (Lines 235-245):

"Lateral boundary conditions were taken from the eddy-resolving Fimbulisen ice shelf–ocean model based on the Regional Ocean Model System (ROMS) [34],

which has been used to investigate basal melting and cavity circulation beneath Fimbulisen under different CDW inflow scenarios that have been observed [35]. The variability of CDW inflow is controlled by the wind stress and hydrographic forcing, with experiment names reflecting combinations of winter or summer hydrography and weak (30), mean (100), and strong (130) wind forcing. From the 18 scenarios, we selected the WIN-100 and SUM-30 climatologies as lateral boundary conditions for the COLD and WARM forcing experiments, respectively. WIN-100 represents a scenario with a homogeneous WW-filled cavity and no CDW inflow, and SUM-30 represents a scenario with constant CDW inflow to the cavity, resulting the highest melt rates in the deep-ice region. All simulations were initialized from the WIN-100 climatology. In the COLD forcing experiments, cavity conditions consistent with WIN-100 hydrography and circulation were maintained through the lateral open-boundary conditions. In the WARM experiments, the cavity transitions from the initial COLD state to WARM conditions through lateral open-boundary forcing derived from the SUM-30 climatology."

Reviewer #3

This paper investigates the impact of channelised basal topography and ocean conditions on both ice-shelf cavity circulation and melt rates on the Jutulstraumen ice stream. The paper presents interesting results which advance our knowledge of the role basal channels play in Antarctic ice shelf stability. Notably, the study focuses on a cold-water ice-shelf cavity, which unveils new circulation dynamics driven by basal topography. Furthermore, the study adds evidence that channelised topography could increase ice-shelf integrated melt rates. In general, the quality of the manuscript and the results shown are consistent with publication in Nature Communications. However, the following comments should be addressed before the manuscript can be recommended for publication.

We thank the reviewer for the thoughtful and supportive feedback and useful suggestions that helped us to improve our manuscript. Below, we respond to each comment in turn, with our responses indicated in blue.

- In the abstract, the authors claim that basal channels ‘focus thinning and reduce stability’. I believe that the overall role of channels on ice shelf stability is still a subject of debate, and so I would be careful making such a definitive statement.

We agree with the reviewer that the overall role of basal channels in ice-shelf stability remains a subject of debate. Basal channels may stabilize ice shelves by limiting area-wide basal melting (Gladish et al., 2012; Millgate et al., 2013). Alternatively, enhanced bottom crevassing and eventual channel melt-out may structurally weaken ice shelves (Alley et al., 2022). To reflect this context dependence, we have revised the abstract to avoid a definitive statement about ice-shelf stability and instead emphasize the role of basal channels in modulating basal melt rates and spatial patterns, replacing the phrase “*focus thinning and reduce stability*” with “*modulate ice-shelf basal melt rates and influence ice-shelf stability*”.

- At the end of the abstract, the authors claim their results show that ‘even modest CDW intrusions could destabilize cold Antarctic ice shelves’. The authors do show that the presence of basal channels increases the overall melt response to CDW intrusions from a 21% increase to a 35% increase. However, I don’t think this is evidence that it could destabilise cold Antarctic ice shelves.

We agree with the reviewer that our simulations do not directly demonstrate ice-shelf destabilization, but rather show that the sensitivity of the ice shelf basal mass loss to ocean warming is enhanced in the presence of basal channels. To avoid overstatement, we have replaced the phrase “*could destabilize cold Antarctic ice shelves*” with “*could have important implications for the stability of cold Antarctic ice shelves*”.

- The results presented show that introducing basal channels increases the overall

ice-shelf integrated melt rate. As mentioned by the authors in the final paragraph lines 155-157), this opposes earlier idealised work. I think including this current state of knowledge in the introduction would help shape the reader's understanding of the significance of these results.

We agree with the reviewer that placing our results in the context of existing work is important for shaping the reader's understanding of their significance. While the original Introduction discussed studies reporting destabilizing effects of basal channels, it did not explicitly highlight the contrasting conclusions drawn from earlier idealized modeling studies. We have therefore revised the Introduction to include this complementary perspective, noting that some idealized studies suggest reduced area-wide basal melting and potential stabilization associated with basal channels.

Specifically, Lines 25-26 in the original manuscript have been revised to the following sentences (Lines 25-30 of the revised version):

"By redistributing basal melting, basal channels influence ice-shelf basal melt rates and spatial patterns, an important factor for ice-shelf stability. Despite increasing observational and modeling efforts, it remains unclear whether basal channels stabilize or destabilize Antarctic ice shelves. The former could be facilitated by preventing area-wide basal melting, leading to a net strengthening of ice shelves [12,13]. Alternatively, basal channels concentrate melting [14] and promote structural weakening [10, 15], potentially leading to ice-shelf destabilization [16]."

- In lines 30-31, the authors state that 'From an oceanic perspective, in the absence of these external factors, sustaining channelized morphology requires differential melting, ...'. If we are in the absence of all other factors, uniform melting would sustain a channel. If we are including ice's secondary flow, then the authors are right. Please clarify this statement.

We thank the reviewer for pointing this out. We agree that the original wording could be ambiguous regarding which processes are excluded. We have therefore clarified the statement by explicitly specifying the external factors being referred to and by explicitly accounting for ice-creep-driven channel closure.

Specifically, Lines 30-31 in the original manuscript have been revised to the following sentences (Lines 33-35 of the revised version):

"From an oceanic perspective, in the absence of upstream ice-dynamical or subglacial forcing, sustaining channelized morphology against ice creep closure [23,24,25] requires differential melting,"

- I am intrigued by the vertical resolution of the model used. If the model has 24 terrain following vertical layers, then the resolution depends on the water column thickness. By introducing a channel, you are increasing the local water column thickness and, therefore, presumably coarsening the vertical resolution. How do the authors think this will affect their results? Could this be contributing to the difference in melt seen between the SMOOTH and ROUGH ice base? It might be nice to see an attempt at making this more comparable between the different runs, perhaps by introducing more vertical layers in the ROUGH runs such that the average vertical resolution in the cavity is similar between the SMOOTH and ROUGH runs.

We appreciate that the reviewer raised this issue. It is correct that introducing a channel changes the local water-column thickness, which in turn modifies the local vertical spacing of sigma layers. This affects the sampling depth of the surface-layer temperature used in the melt parameterization, but it does not alter the underlying circulation or water masses. In our model setup, introducing a channel increases the local water-column thickness at the channel apex and decreases it at the channel base when the same number of vertical layers is used. In specific, as Figure 1 shows, the example channel apex (base), the water-column thickness increases (decreases) 70.2 m (97.2 m) relative to the water thickness of 554.1 m (551.2 m) in the SMOOTH ice base, contributing to a 12.7% (17.4%) decrease (increase) in vertical resolution in each layer.

Because sigma layers are terrain-following and distributed proportionally to the total water depth, the sampling depth of the surface-layer temperature moves 12.7% (17.4%) downward (upward) as the water column becomes deeper (shallower). This can slightly modify the local melt rate depending on the vertical temperature gradient. However, in our experiments the temperature structure in the upper layers is generally homogeneous under both forcing conditions (Figure 2), so the magnitude of this effect is small. In contrast, the melt differences between the ROUGH and SMOOTH cases arise primarily from circulation changes due to the existence of the channel geometry, not from differences in sampling depth, as we elaborated in the manuscript. In order to make this point clearer, we add the following paragraph in the Methods in the revised manuscript (Lines 251-253):

"Although the local vertical spacing of sigma layers under the ROUGH and SMOOTH drafts differs, the resulting changes in surface-layer sampling depth have only a minor influence on basal melt rates in our simulations, which are instead dominated by circulation changes associated with the channel geometry."

Regarding the reviewer's suggestion to repeat the experiment with identical vertical resolution, we note that modifying the number of sigma layers would only change the thickness of the top layer (sampling depth) without affecting the circulation, stratification, or water masses. Furthermore, because introducing a channel modifies the water-column thickness in opposite directions at the chan-

nel apex and base, it is not possible to make the vertical resolution identical everywhere. For these reasons, we do not expect such a sensitivity test to alter the conclusions of this study.

Figure 1: **Cross-sections of sigma-layer structure at the example channel under ROUGH and SMOOTH ice geometries.** The blue thick line shows the ice draft for the channel, and the blue thin lines show the corresponding sigma-layer depths. For the smoothed version of the channel, the ice draft is shown as a grey thick dashed line and the sigma-layer depths as grey thin dashed lines. The black thick line marks the ocean bottom.

Figure 2: **Temperature cross-sections of the example basal channel under different forcing and ice-draft geometries.** **a, b,** Temperature cross-sections of the smoothed channel geometry under COLD (a) and WARM (b) forcing. **c, d,** Temperature cross-sections of the channel geometry under COLD (c) and WARM (d) forcing. Gray contours mark sigma-layer depths for corresponding ice cavity geometries.

- *The authors show that the increase in ‘area integrated melt rates when changing from COLD to WARM forcing is four times larger under the ROUGH ice draft (0.4 m/yr) compared with the SMOOTH ice draft (0.1 m/yr)’. This is a really interesting result! However, the authors need to exert more care when using the term ‘fourfold’. The use of the current statistic on line 153 is particularly overstated – ‘In this way, even modest CDW incursions can trigger a fourfold increase in ice mass loss below deep ice. . .’. Firstly, as the study doesn’t include the effects of melt on ice dynamics, I think ‘mass loss’ here should be replaced with ‘melt rate’. This is the same in the abstract (line 16). While saying there has been a ‘fourfold increase’ is technically correct for the absolute values in the runs presented, making general claims using this statistic is misleading for the reader, as the baselines of the two cases are different. It would be beneficial if the authors made this clear within the text to avoid confusion. When discussing the result in a more general sense, it might worth considering using percentage change instead.*

We agree with the reviewer that using the term “fourfold” can be misleading when comparing cases with different baseline melt rates, and that greater care is required in presenting this result. Following this suggestion, we have removed the term “fourfold increase” from both the abstract and the main text to avoid overstatement.

We also agree that the term “ice mass loss” is not appropriate in this context, as it typically refers to dynamic mass loss of the ice sheet across the grounding line in response to ice-shelf thinning. Accordingly, we have replaced “ice mass loss” with “basal mass loss” in the abstract, referring specifically to ice loss from the underside of the ice shelf due to basal melting. In the main text, we have replaced “ice mass loss” with “basal melt rates,” which more accurately reflects the quantity analysed.

Specifically, in the abstract we have replaced the original phrase *"The resulting differential melting promotes channel growth and drives a fourfold increase in total basal mass loss, undermine"* with *"This ocean-driven process significantly enhances the sensitivity of the ice shelf basal mass loss to ocean warming, and the resulting differential melting promotes channel growth, with the potential to undermine"*.

Similarly, in Line 153 of the original manuscript we replaced the phrase *"trigger a fourfold increase in ice mass loss"* with *"significantly increase basal melt rates"* (Line 179 of the revised version).

- *While it is clear that the presence of channels leads to an increase in ice shelf integrated melt in this study, the argument would be strengthened if the authors could show that this increase is statistically significant. I suggest using a permutation test for matched pairs (for example, Wilcoxon’s Signed Rank test) to compare the difference between the ROUGH and SMOOTH ice draft in both*

ocean conditions.

Figure 3: **Distributions of paired grid-point melt rate anomalies between ROUGH and SMOOTH drafts under COLD and WARM forcing.** **a, b,** Boxplots of melt rate anomalies for the smoothing region (a) and the Deep-ice region (b). The black horizontal line within each blue box indicates the median anomaly, and the box spans the inter-quartile range (25th–75th percentiles). Red dashed lines denote zero anomaly, corresponding to no difference between ROUGH and SMOOTH drafts. Whiskers extend to the most extreme non-outlier values. P-values from two-sided Wilcoxon signed-rank tests for each experiment are shown above the whiskers.

Thank you for this important suggestion. We have now assessed the difference between ROUGH and SMOOTH draft experiments using a non-parametric Wilcoxon signed rank test applied to paired grid-point basal melt-rate anomalies. The analyses was conducted separately for the smoothing region and the deep-ice region under both COLD and WARM forcing conditions. In all cases, the resulting p-values are smaller than 0.001, indicating statistically significant differences between the ROUGH and SMOOTH draft experiments.

To illustrate the statistical results and their spatial consistency, here we provide boxplots of paired grid-point melt rate anomalies for both regions and forcing conditions (Figure 3). These boxplots show that, in both regions and under both forcing conditions, the median anomaly is positive, demonstrating that the enhanced basal melt under the ROUGH ice draft reflects a spatially consistent signal rather than being driven by isolated extreme values.

We have added the following sentence in the revised manuscript (Lines 88-89) where we compare the area-averaged basal melt rates between the ROUGH and SMOOTH drafts, to explicitly link the melt rate increase to the statistical analysis:

"This increase reflects a spatially consistent enhancement of grid-point melt rates across the ice shelf (Wilcoxon signed-rank test, $p < 0.001$; Methods)."

In addition, in the Methods section of the revised manuscript (Lines 278-283), we have added a corresponding paragraph describing the statistical analysis:

"Statistical analysis"

"Differences between ROUGH and SMOOTH draft experiments were assessed using a non-parametric Wilcoxon signed-rank test applied to paired grid-point basal melt rate anomalies (ROUGH relative to SMOOTH). Analyses were conducted separately for the smoothing region and the deep-ice region under both COLD and WARM forcing. The test evaluates whether the median anomaly differs from zero and is therefore appropriate for identifying spatially consistent signals in the presence of non-Gaussian distributions and spatial heterogeneity. Statistical significance was evaluated using a two-sided test. "

Figure 1: - Please make the green dashed line more clear - In e and f, please indicate which way round the channel is - is the reader looking forwards the grounding line or the calving front?

We have increased the line thickness of the green dashed transect in Figure 1a to improve its visibility. In addition, we revised the figure caption in the revised manuscript to clarify the orientation of the channel cross-section. Specifically, the sentence *" e, f, Temperature cross-sections of the example channel under COLD (e) and WARM (f) forcing."* has been revised to *" e, f, Cross-channel (west to east) temperature sections through the example channel under COLD (e) and WARM (f) forcing."*

Note that this section is taken at a fixed along-flow location, so both sides of the section are equidistant from the grounding line and the ice front.

Figure 2: - Again, please orient the transect used in c-f.

We revised the figure caption in the revised manuscript to clarify the orientation of the channel cross-section. Specifically, the sentence *"The yellow solid line indicates a cross-section of the example channel illustrated in the following panels."* has been revised to *" The yellow solid line indicates the cross-channel (west to east) sections through the example channel illustrated in the following panels."*

Figure 4: - It's currently hard to tell the difference between the 'ice flow' line and the 'Ocean, COLD' line

We have increased the line thickness of the " ice flow" line (blue dashed line) in Figure 4 to improve its visibility.

References

- Alley, K.E., Scambos, T.A., Alley, R.B., 2022. The role of channelized basal melt in ice-shelf stability: recent progress and future priorities. *Ann. Glaciol.* 63, 18–22.
- Gladish, C.V., Holland, D.M., Holland, P.R., Parice, S.F., 2012. Ice-shelf basal channels in a coupled ice/ocean model. *J. Glaciol.* 58, 1527–1544.
- Hattermann, T., Smedsrud, L., Nøst, O., Lilly, J., Galton-Fenzi, B., 2014. Eddy-resolving simulations of the Fimbul Ice Shelf cavity circulation: Basal melting and exchange with open ocean. *Ocean Model.* 82, 28–44.
- Jackett, D.R., McDougall, T.J., 1995. Minimal adjustment of hydrographic profiles to achieve static stability. *Journal of Atmospheric and Oceanic Technology* 12, 381–389.
- McDougall, T.J., Barker, P.M., Feistel, R., Galton-Fenzi, B.K., 2014. Melting of ice and sea ice into seawater and frazil ice formation. *Journal of Physical Oceanography* 44, 1751–1775.
- Millgate, T., Holland, P.R., Jenkins, A., Johnson, H.L., 2013. The effect of basal channels on oceanic ice-shelf melting. *J. Geophys. Res.* 118, 1–14.
- Wearing, M., Stevens, L., Dutrieux, P., Kingslake, J., 2021. Ice-shelf basal melt channels stabilized by secondary flow. *Geophys. Res. Lett.* 48, e2021GL094872.

Reviewer #2

I would like to thank the authors for their thoughtful responses to my major comments. I noticed that only the major comments were addressed, and the minor comments were missing. I apologise if my minor comments went missing from my initial assessment, but I would like to reiterate them here for the authors' consideration. I have updated them to only keep those pertinent after their first review. Overall, I think the authors have made significant improvements to the manuscript, and I would like to recommend for publication after minor revisions.

We thank the reviewer for the positive evaluation of our revisions and for reiterating the minor comments. We apologize for inadvertently overlooking these comments in our previous response. We have now carefully addressed each of them below and revised the manuscript where appropriate.

Below, we respond to each comment in turn, with our responses indicated in blue. Line numbers referring to the "original manuscript" correspond to the revised version submitted after the first round of review, unless otherwise stated.

Minor Comments

-[Line 86] It will be useful for the reader to add the percentage of the melt rate increase between simulations in a sentence here.

We thank the reviewer for this suggestion. The percentage increase in basal melt rates has now been included in the revised manuscript. Specifically, Lines 86–88 in the original manuscript have been revised as follows (Lines 89–91 in the revised version):

"Consistently, the area-averaged basal melt rates are higher under the ROUGH ice draft than under the SMOOTH ice draft, increasing by 18% (27%) in the smoothing region and 21% (35%) in the deep-ice region under COLD (WARM) forcing conditions (Figure 1b, Methods)."

- [Line 96] I find this line confusing, consider rephrasing it. I would suggest saying: "In the deep-ice region (near the grounding line), the difference in the mean melt rate between the smooth warm and cold cases is small (0.1 m/yr), whereas in the rough warm and cold cases, the mean melt rate difference is 0.4 m/yr (4 times larger)."

Thank you for this suggestion. We have rephrased the sentence to clarify the comparison between the COLD and WARM forcing cases while retaining the original meaning. Specifically, Lines 96–97 in the original manuscript have been revised as follows (Lines 99–100 in the revised version):

"Especially in the deep-ice region, the increase in area-averaged melt rates from COLD to WARM forcing is four times larger under the ROUGH ice draft

(0.4 m yr⁻¹) than under the SMOOTH ice draft (0.1 m yr⁻¹)."

- [Figure 2] Consider adjusting the colorbar in panel c, I am surprised that there is no salinity signature of melt water, despite the fact that the left flank of the channel is experiencing melting.

We thank the reviewer for this suggestion. As we explained in the earlier response to Reviewer #1, Figure 2c shows the reference salinity, which reflects the source water mass modified by meltwater mixing along the Gade line. Under COLD forcing, the water within the channel is dominated by Winter Water and therefore exhibits a nearly uniform reference salinity of around 34.35. As a result, adjusting the colorbar range would not reveal additional spatial structure in this panel. However, we note that temperature variations within the channel indicating the presence of meltwater are shown in Figure 1e.

- [Line 103] Emphasise here the role of the velocity field and it's resemblance to the melt rates. This is linked to my previous major comment.

Thank you for this comment. As noted in our response to the reviewer's previous major comment, we have revised the discussion to better emphasize the role of friction velocity and its combined influence with thermal driving on melt-rate anomalies. The sentence here already highlights that turbulent heat transfer in the melt parameterization scales with ocean velocity near the ice base, and therefore links the flow speed anomalies to the positive melt-rate anomalies along the western flank of the channel. We believe that, together with the revisions described above, the role of the velocity field and its resemblance to the melt-rate patterns are now sufficiently clear in the manuscript.

-[Line 114] Add space between Figure and 2c.

Added.

- [Line 120] The analysis of the water mass properties is very interesting and an excellent addition to the manuscript.

Thanks for the very positive comment.

- [Line 204] Add space between 1°C and at.

Added.

- [Line 228] Add space between Figure and 1d.

Added.

- [Line 159] Add space between observations and such.

Added.

- [Line 160] Add space between [49] and can.

Added.